# Lithium carbonate-promoted mixed rare earth oxides as a generalized strategy for oxidative coupling of methane with exceptional yields

Kun Zhao[1,2], Yunfei Gao ®[3] ✉, Xijun Wang[4], Bar Mosevitzky Lis ®[5], Junchen Liu[1], Baitang Jin ®[1], Jacob Smith[1], Chuande Huang[6], Wenpei Gao ®[1], Xiaodong Wang ®[6], Xin Wang[3], Anqing Zheng[2], Zhen Huang[2], Jianli Hu ®[7], Reinhard Schömacker ®[8], Israel E. Wachs ®[5] ✉ & Fanxing Li ®[1] ✉

The oxidative coupling of methane to higher hydrocarbons offers a promising autothermal approach for direct methane conversion, but its progress has been hindered by yield limitations, high temperature requirements, and performance penalties at practical methane partial pressures (~1 atm). In this study, we report a class of $Li_2CO_3$-coated mixed rare earth oxides as highly effective redox catalysts for oxidative coupling of methane under a chemical looping scheme. This catalyst achieves a single-pass $C_{2+}$ yield up to 30.6%, demonstrating stable performance at 700 °C and methane partial pressures up to 1.4 atm. In-situ characterizations and quantum chemistry calculations provide insights into the distinct roles of the mixed oxide core and $Li_2CO_3$ shell, as well as the interplay between the Pr oxidation state and active peroxide formation upon $Li_2CO_3$ coating. Furthermore, we establish a generalized correlation between $Pr^{4+}$ content in the mixed lanthanide oxide and hydrocarbons yield, offering a valuable optimization strategy for this class of oxidative coupling of methane redox catalysts.

Efficient, single-step conversion of methane into value-added chemicals has been a critical challenge in C1 chemistry. Among the various conversion methods, oxidative coupling of methane (OCM), which employs gas-phase molecular $O_2$ to generate higher hydrocarbons ($C_{2+}$) in an autothermal process, has garnered significant research attention since its inception in the 1980s[1]. Over the past 40 years, ~2000 OCM catalysts have been identified for OCM through

experimental screening and/or with the assistance of machine learning[1–4]. Among the investigated materials, top-performing candidates predominantly fall within two distinct catalyst families: the unsupported Li-MgO mixed oxide and the supported Mn-$Na_2WO_4$/$SiO_2$.

The Li-MgO bulk mixed oxide catalyst was first reported by Lunsford et al. in 1985 and achieved up to 19% $C_{2+}$ yield at 720 °C[5,6].

[1]North Carolina State University, Campus Box 7905, Raleigh, NC, USA. [2]CAS Key Laboratory of Renewable Energy, Guangdong Provincial Key Laboratory of New and Renewable Energy Research and Development, Guangzhou Institute of Energy Conversion, Chinese Academy of Sciences, Guangzhou, China. [3]Institute of Clean Coal Technology, East China University of Science and Technology, Shanghai, China. [4]Department of Chemical and Biological Engineering, Northwestern University, Evanston, IL, USA. [5]Operando Molecular Spectroscopy & Catalysis Laboratory, Department of Chemical & Biomolecular Engineering, Lehigh University, Bethlehem, PA, USA. [6]Dalian Institute of Chemical Physics, Chinese Academy of Sciences, Dalian, China. [7]Department of Chemical & Biomedical Engineering, West Virginia University, Morgantown, WV, USA. [8]Department of Chemistry, Technische Universität Berlin, Straße des 17. Juni 124, Berlin, Germany. ✉e-mail: yunfeigao@ecust.edu.cn; iew0@lehigh.edu; fli5@ncsu.edu

While various optimizations have been conducted on this catalyst, the $C_{2+}$ yield has yet to exceed 20%[7,8]. Moreover, catalyst deactivation issues persist due to evaporation of lithium in the form of LiOH[9,10]. In 1992, Fang et al. reported that the supported Mn-$Na_2WO_4$/$SiO_2$ catalyst demonstrated 23.9% $C_{2+}$ yield at 800 °C[11]. Extensive studies were performed on this catalyst including material screening[12], surface and bulk structural characterization[13], reaction pathway and mechanism modeling[14], and reactor optimization[15]. While deeper mechanistic insights into these catalyst families have been obtained in recent years[14,16,17], the observed $C_{2+}$ yield has not exceeded 30% for the Mn-$Na_2WO_4$/$SiO_2$ catalyst family, which showed satisfactory stability in general. Outside of the Li-MgO and Mn-$Na_2WO_4$/$SiO_2$ catalyst families, $La_2O_3$-$CeO_2$ nanofibers have exhibited a $C_{2+}$ yield of ~20% at the relatively low temperature of 520 °C[18]. By maximizing all the desired reaction rates and optimizing thermochemistry for all the surface species on an idealized catalyst, Green et al. predicted, through kinetic modeling, that the $C_{2+}$ yield would limit to ~28% in catalytic OCM with $O_2$-cofeed[19]. This is consistent with the experimentally reported yields to date.

To address the yield limitations from co-feeding methane and gaseous $O_2$ for OCM. Up to 34.7% $C_{2+}$ yield was reported at 900 °C in diluted methane ($P_{CH_4} < 0.5$ atm) using a catalytic membrane reactor composed of a mixed-conductive $Ba_{0.5}Ce_{0.4}Gd_{0.1}Co_{0.8}Fe_{0.2}O_{3-\delta}$ membrane and a supported Mn-$Na_2WO_4$/$SiO_2$ catalyst. However, rapid membrane degradation was observed, a common challenge for membrane-based OCM at such elevated temperatures[20]. Lattice oxygen-based OCM has also been performed under a chemical-looping (CL) mode, which utilizes a reducible metal oxide operated through cyclic redox steps under alternating methane and $O_2$ environments[21,22]. Gaffney et al. pioneered the concept of chemical looping-OCM (CL-OCM) and reported that a Na impregnated $Pr_6O_{11}$ catalyst achieved up to 16% $C_{2+}$ yield at 775 °C[23]. The chemical looping mode utilizes the redox between $Pr^{4+}$ and $Pr^{3+}$ and demonstrated ~4% higher $C_{2+}$ yield than the $O_2$-cofeed mode. More recently, Fan et al. proposed the idea of CL-OCM by designing a Li and W co-doped $Mg_6MnO_8$ redox catalyst that exhibited 28.6% $C_{2+}$ yield at 850 °C[24,25]. To date, more than 10,000 articles have been published on OCM. However, none of the prior studies have demonstrated >30% $C_{2+}$ yield with satisfactory stability. Moreover, most of these studies were carried out with highly diluted methane, which would not be suitable for practical applications. Based on experimental data coupled with kinetic analyses, Labinger et al. argued that higher methane partial pressures would lead to severe yield penalties[26]. On the other hand, it has been estimated that for OCM to achieve commercial viability, a $C_{2+}$ yield exceeding 30–35% at practical partial pressures (~1 atm) is required[27]. As such, a gap clearly exists between reported academic research results and industrial application[28].

From a mechanistic aspect, various active sites or active species have been postulated to be responsible for methane activation. Taking Li-MgO as a model catalyst, early studies by Lunsford et al. suggested that $Li^+O^-$ is the active site based on electron spin resonance (EPR) of quenched catalysts in the presence of $O_2$ with the $g_\perp = 2.054$ signal[5,29]. Based on the O 1$s$ shoulder observed at 533 eV in ex-situ X-ray photoelectron spectroscopy (XPS) using a spectrometer equipped with a pretreatment chamber, Stair et al. argued for the presence of peroxide or $Li^+O^-$ species in the surface region (<3 nm)[30]. The presence of peroxides for OCM on a related, unsupported Ba/MgO mixed oxide catalyst was reported by Lunsford et al. on the basis of in-situ Raman (with the $BaO_2$ band at 842 $cm^{-1}$) and ex-situ XPS (with the O 1$s$ peak at 531 eV)[31,32]. While the presence of peroxides was largely confirmed, the necessity for $Li^+O^-$ sites was questioned by subsequent studies involving both experimental work and quantum chemistry calculations[33,34]. These studies argued that $Mg^{2+}O^{2-}$ sites or defective MgO surfaces are responsible for methane activation, and Li only acts as a structural modifier instead of an active center[33,34]. The search for active sites in the supported Mn-$Na_2WO_4$/$SiO_2$ was similarly challenging. A number of earlier studies, mostly through ex-situ measurements, proposed that the active sites are either Na-O-Mn, Na-O-W or other bonds belonging to bulk crystalline phases selected from the Mn-Na-W-O components[35]. More recently, Wachs et al. conducted in-situ Raman studies and demonstrated that none of the abovementioned crystalline phases are actually present at the OCM reaction temperature (900 °C), and the active site for methane activation are isolated, pseudotetrahedral Na-coordinated $WO_4$ surface sites (Na-$WO_4$) on the $SiO_2$ support[13,14,16]. Through separate studies, Takanabe and Tao et al. detected the presence of peroxide species for both supported $K_2WO_4$/$SiO_2$ and $Na_2WO_4$/$SiO_2$[36,37] catalysts with in-situ XPS. Using laser induced fluorescence (LIF) measurements, Tao further proposed that the presence of near-surface peroxides can lead to the formation of hydroxyl radicals for methane activation.

Given the potential role of surface/subsurface peroxide species and the redox properties of praseodymium oxides in the context of chemical looping[23], the current study focuses on Pr-containing lanthanide oxides with a $Li_2CO_3$ promoter for CL-OCM. $Li_2CO_3$ was selected because it has good $O_2^{2-}$ solubility and conductivity, and was previously reported to be effective for ethane activation[38]. Unsupported bulk mixed oxides containing Pr and another lanthanide cation, on the other hand, can beneficially modify the redox properties of $Pr^{4+}$/$Pr^{3+}$ [23]. In the present study, a series of Pr-containing lanthanide oxides with a thin surface film of $Li_2CO_3$ ($LnPrO_{3+x}@Li_2CO_3$, Ln = La, Eu, Ho, Dy, Sm, and Nd) for CL-OCM were synthesized and characterized. This family of materials exhibited up to 30.6% single-pass $C_{2+}$ yield with stable performance at 700 °C. The roles of the mixed oxide core and $Li_2CO_3$ shell, as well as the interplays among the Pr oxidation state, active peroxide formation upon $Li_2CO_3$ coating, and OCM performance were determined by ex-situ X-ray absorption near edge structure (XANES), in-situ Raman, in-situ X-ray diffraction (XRD), in-situ XPS, and quantum chemistry calculations.

## Results

### Structures of catalyst bulk phase and surface region under different environments

While all the $Li_2CO_3$ promoted $LnPrO_{3+x}$ oxides (Ln = La, Eu, Ho, Dy, Sm, Nd) were active for OCM (as will be discussed in later sections), $LaPrO_{3+x}@5Li_2CO_3$ (5 refers to 5 wt.% $Li_2CO_3$ loading) was selected as a representative redox catalyst for detailed characterizations since it showed excellent performance and La is a relatively abundant rare earth element. The $LaPrO_{3+x}@5Li_2CO_3$ redox catalyst consists of a core-shell structure. The core consists of the crystalline $LaPrO_{3+x}$ bulk phase as shown by in-situ XRD at 700 °C (Fig. 1a). The crystalline $LaPrO_{3+x}$ core is covered by a thin $Li_2CO_3$ shell as revealed by (i) spatial distribution of Li in the mapping of the ex-situ TEM-EELS analysis on a catalyst particle (Fig. 1b), (ii) surface enrichment of carbon with TEM-EDS (Supplementary Fig. S1) whereas carbonate-free $LaPrO_{3+x}$ does not exhibit a XPS signal for carbon (Fig. 1c), (iii) presence of carbon and lithium in the surface region of $LaPrO_{3+x}@5Li_2CO_3$ with in-situ XPS (Supplementary Fig. S2), and (iv) absence of Pr and La on the outermost surface layer (0.3 nm) as revealed by high sensitivity - low energy ion scattering (HS-LEIS) analysis of the surface, and the increase in the La and Pr signals with sputtering depth (Fig. 1d). The thin $Li_2CO_3$ shell (<5 nm) is amorphous and lacks long range order. Therefore, its signal does not appear in the in-situ XRD pattern (Fig. 1a). Furthermore, bulk $Li_2CO_3$ melts at 723 °C, suggesting that the surface layer is likely be in a molten state under the OCM reaction conditions (~700 °C) given the lower melting temperatures of thin films[38]. The in-situ TEM analysis (Fig. 1e) further verifies the morphology and composition of the $LaPrO_{3+x}@5Li_2CO_3$ catalyst, which consists of a crystalline $LaPrO_{3+x}$ core enveloped by a thin amorphous $Li_2CO_3$ shell at 700 °C.

## The relationship between near-surface peroxide and Pr⁴⁺

Given that CL-OCM reactions proceed through cyclic removal (OCM step) and replenishment (re-oxidation step) of lattice oxygen from bulk reducible oxides in the redox catalyst, the dynamics of the bulk $LaPrO_{3+x}$ phase for $LaPrO_{3+x}$ and $LaPrO_{3+x}@5Li_2CO_3$ under oxidizing and methane reducing conditions were further monitored with in-situ XRD, Raman and XPS at 700 °C. The oxidized Li-free bulk $LaPrO_{3+x}$ mixed oxide is present as cubic-$LaPrO_{3.33}$ and transforms to a mixture of cubic-$La_2O_3$ and cubic-$Pr_2O_3$ after methane reduction (see in-situ XRD in Supplementary Fig. S3). This is corroborated by the corresponding in-situ Raman spectra (Fig. 2a, b), showing that the oxidized cubic-$LaPrO_{3.33}$ phase (572 cm⁻¹) is reduced to cubic-$La_2O_3$ and cubic $Pr_2O_3$ (112 and 302 cm⁻¹). Westermann et al. assigned the band at

572 cm⁻¹ to Pr⁴⁺ defects[39]. Re-oxidation converts the reduced phase back to the initial oxidized state. Cubic-$LaPrO_{3+x}$ is also present in the oxidized $LaPrO_{3+x}@5Li_2CO_3$ catalyst, but it reversibly transforms to the bulk hexagonal-$LaPrO_{3+x}$ phase after methane reduction. This is confirmed by in-situ XRD in Supplementary Fig. S3 and in-situ Raman spectra in Fig. 2c, d, where the oxidized bulk c-$LaPrO_{3+x}$ (572 cm⁻¹) was reduced to the bulk h-$LaPrO_{3+x}$ (178 and 392 cm⁻¹). Thus, the amorphous $Li_2CO_3$ shell affects the structure of the bulk $LaPrO_{3+x}$ phase under the OCM conditions. The presence of the $Li_2CO_3$ shell also resulted in the formation of peroxide species ($O_2^{2-}$: in-situ Raman band at ~850 cm⁻¹ characteristic of $Li_2O_2$)[40] during the transient oxidation of the reduced $LaPrO_{3+x}@5Li_2CO_3$ mixed oxide. The absence of peroxide species for the Li-free $LaPrO_{3+x}$ and the presence of the peroxide

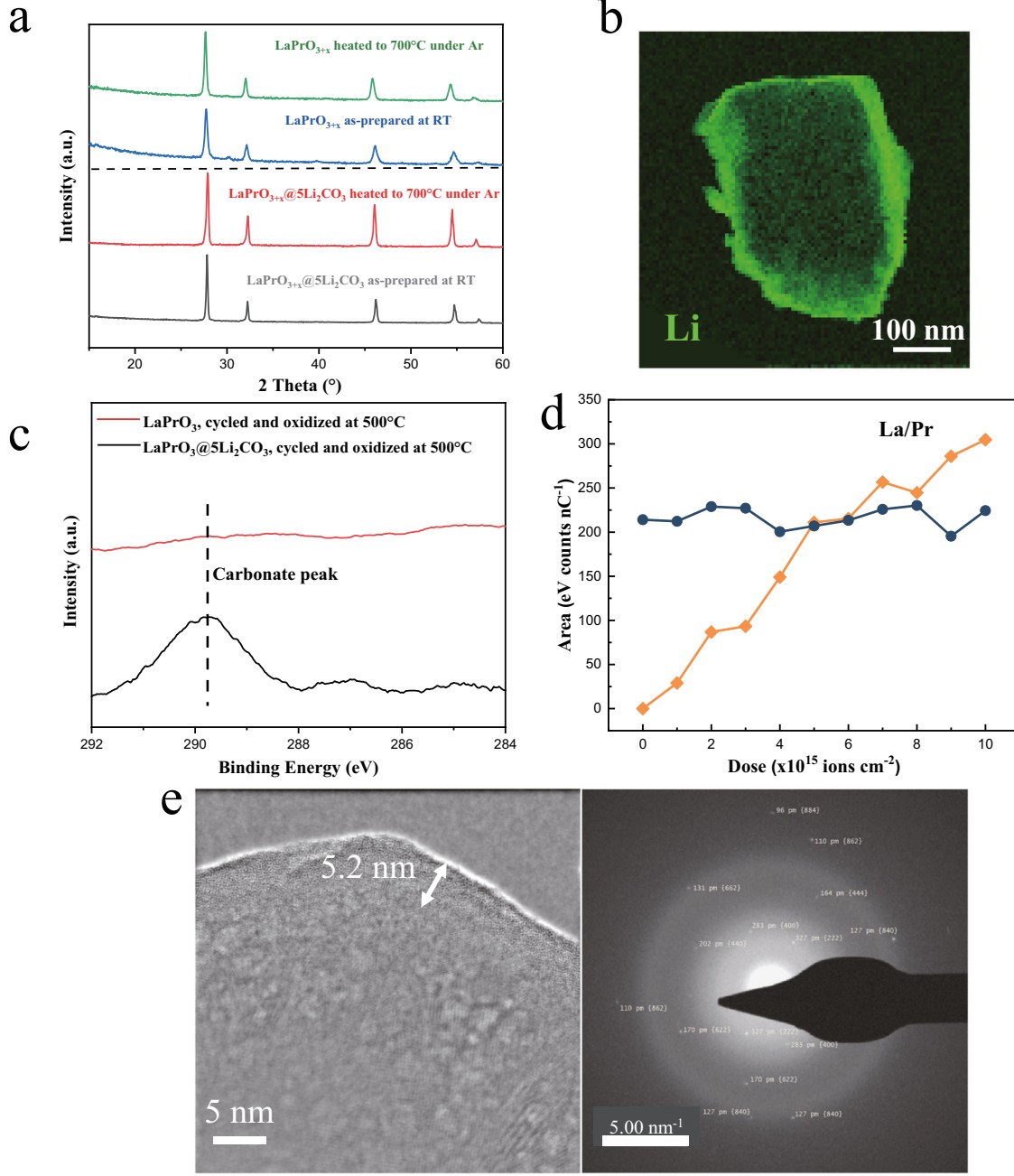

**Fig. 1 | Ex-situ and in-situ spectroscopic characterizations for $LaPrO_{3+x}@5Li_2CO_3$. a** In-situ XRD on $LaPrO_{3+x}@5Li_2CO_3$ under air at 700 °C; **b** Ex-situ TEM-EELS on $LaPrO_{3+x}@5Li_2CO_3$ in vacuum; **c** In-situ XPS on $LaPrO_{3+x}$ and $LaPrO_{3+x}@5Li_2CO_3$, both $LaPrO_{3+x}$ and $LaPrO_{3+x}@5Li_2CO_3$ were reduced with diluted methane at 700 °C and re-oxidized with diluted oxygen at 500 °C in the in-situ XPS chamber; **d** Quasi in-situ HS-LEIS on $LaPrO_{3+x}@5Li_2CO_3$ treated in 600 °C under 10% $O_2$ in a pretreatment chamber; **e** In-situ TEM and electron diffraction of $LaPrO_{3+x}@5Li_2CO_3$ at 700 °C under diluted $O_2$.

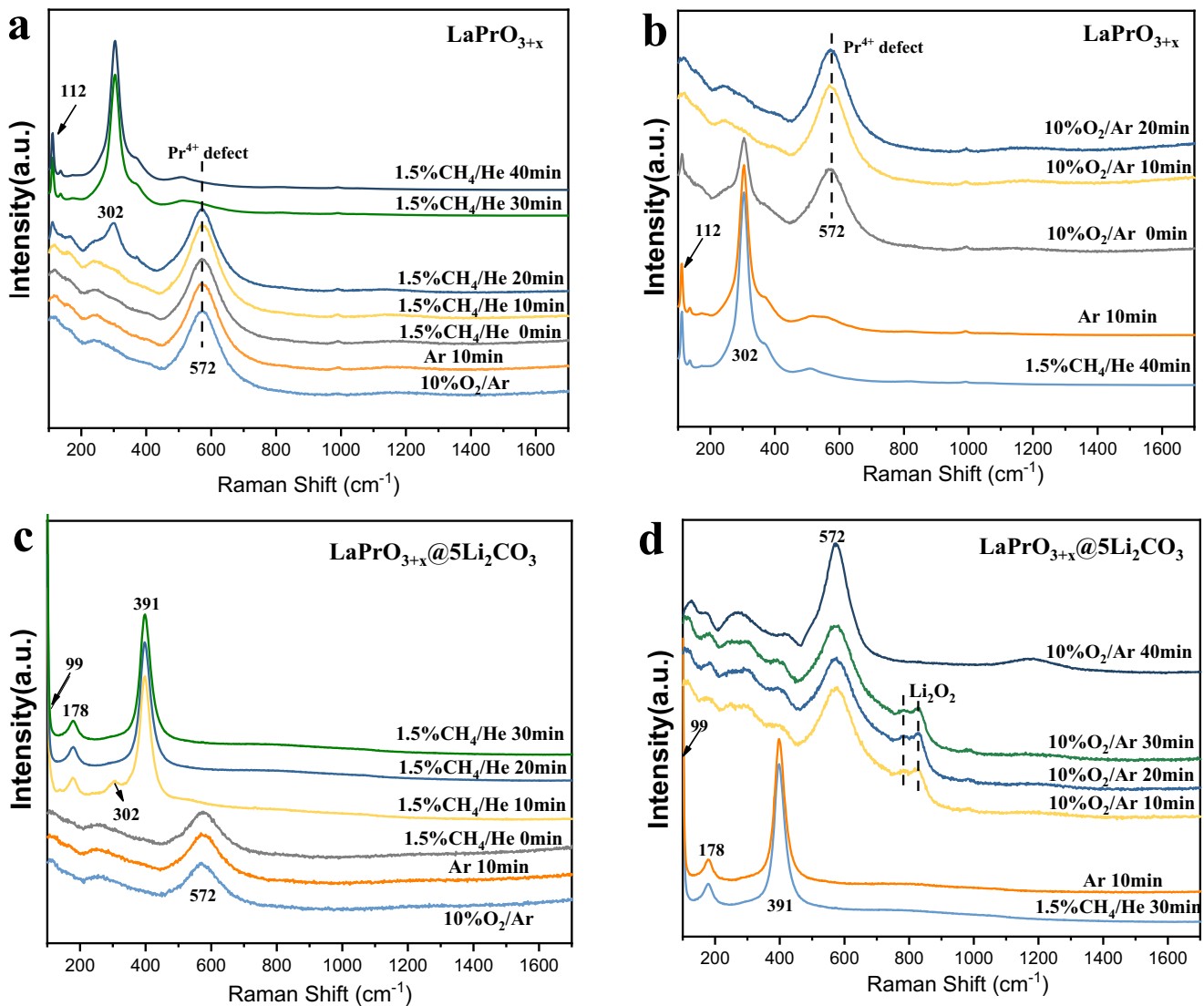

**Fig. 2 | In-situ Raman for LaPrO$_{3+x}$ and LaPrO$_{3+x}$@5Li$_2$CO$_3$.** In-situ Raman for **a** LaPrO$_{3+x}$ reduction at 700 °C, **b** LaPrO$_{3+x}$ reoxidation at 700 °C, **c** LaPrO$_{3+x}$@5Li$_2$CO$_3$ reduction at 700 °C, **d** LaPrO$_{3+x}$@5Li$_2$CO$_3$ reoxidation at 700 °C.

species for LaPrO$_{3+x}$@5Li$_2$CO$_3$ suggest that the peroxide species are associated with the thin Li$_2$CO$_3$ shell. It is noted that the surface area of Li$_2$CO$_3$ promoted LaPrO$_{3+x}$ is quite low (~1 m$^2$/g, Supplementary Table S1 summarizes the surface areas of bare LaPrO$_{3+x}$ and LaPrO$_{3+x}$ with different Li$_2$CO$_3$ loadings), the ability for Raman to detect the peroxide species in this low surface area assay suggests that the detected peroxide signal cannot just be surface bound. Rather, contributions from bulk peroxide species, e.g. peroxides dissolved/incorporated in the amorphous Li$_2$CO$_3$ shell, is more likely. We also note that Raman did not detect Li$_2$CO$_3$ peaks from this sample, this is probably due to its low loading (5 wt.%) and the peak broadening effect of amorphous carbonate as the temperature increases[41]. This peak broadening effect was confirmed via an in-situ Raman experiment on LaPrO$_{3+x}$@5Li$_2$CO$_3$ under 5%CO$_2$ (balance Ar) with temperature ramping up from 120 to 700 °C. As shown in Supplementary Fig. S4a, LaPrO$_{3+x}$@5Li$_2$CO$_3$ exhibited a clear surface carbonate peak between 1100–1300 cm$^{-1}$. This peak, however, tends to be broadened and smoothed out when the temperature gradually ramped up to 700 °C. We note that this broadening effect is not likely due to the thermal decomposition of Li$_2$CO$_3$ since the presence of 5 vol.% CO$_2$ would inhibit carbonate decomposition from a thermodynamic standpoint. We have also compared ex-situ Raman under air in room temperature

for LaPrO$_{3+x}$@3Li$_2$CO$_3$, LaPrO$_{3+x}$@5Li$_2$CO$_3$ and LaPrO$_{3+x}$@10Li$_2$CO$_3$. All these samples exhibited surface carbonate peaks of similar relative intensities (Supplementary Fig. S4b). Thus, the absence of surface carbonate peaks for LaPrO$_{3+x}$@5Li$_2$CO$_3$ under in-situ Raman is more likely due to temperature effect rather than the Li$_2$CO$_3$ loading effect.

Our previous study on chemical looping ethane conversion indicated that the formation of peroxide species from mixed oxides can be linked to the presence of highly reducible cation components[38]. Therefore, the oxidation states of the Pr cations in the oxidized LaPrO$_{3+x}$ and LaPrO$_{3+x}$@5Li$_2$CO$_3$ mixed oxide catalysts were determined by ex-situ XANES first. It was shown that both Pr$^{3+}$ and Pr$^{4+}$ are present for the LaPrO$_{3+x}$ and LaPrO$_{3+x}$@5Li$_2$CO$_3$ catalysts (Supplementary Fig. S5)[42]. It is evident that LaPrO$_{3+x}$@5Li$_2$CO$_3$ contains more bulk Pr$^{4+}$ than LaPrO$_{3+x}$. The corresponding in-situ XPS measurement also detected the presence of near-surface Pr$^{4+}$ and peroxide species. Figure 3a shows the Pr 3d XPS spectra of LaPrO$_{3+x}$ and LaPrO$_{3+x}$@5Li$_2$CO$_3$. Although the quantification of Pr$^{4+}$/Pr$^{3+}$ with XPS is complex, Sinev et al. have reported the characteristic peaks and features for Pr$^{4+}$ with in-situ XPS by switching between oxidizing and reducing atmospheres on a Pr-Ce mixed oxide[43]. A large "a/b" peak area ratio and a large "c" peak area are representative of the Pr$^{4+}$ features (as labeled in Fig. 3a). As shown in Fig. 3a, LaPrO$_{3+x}$@5Li$_2$CO$_3$

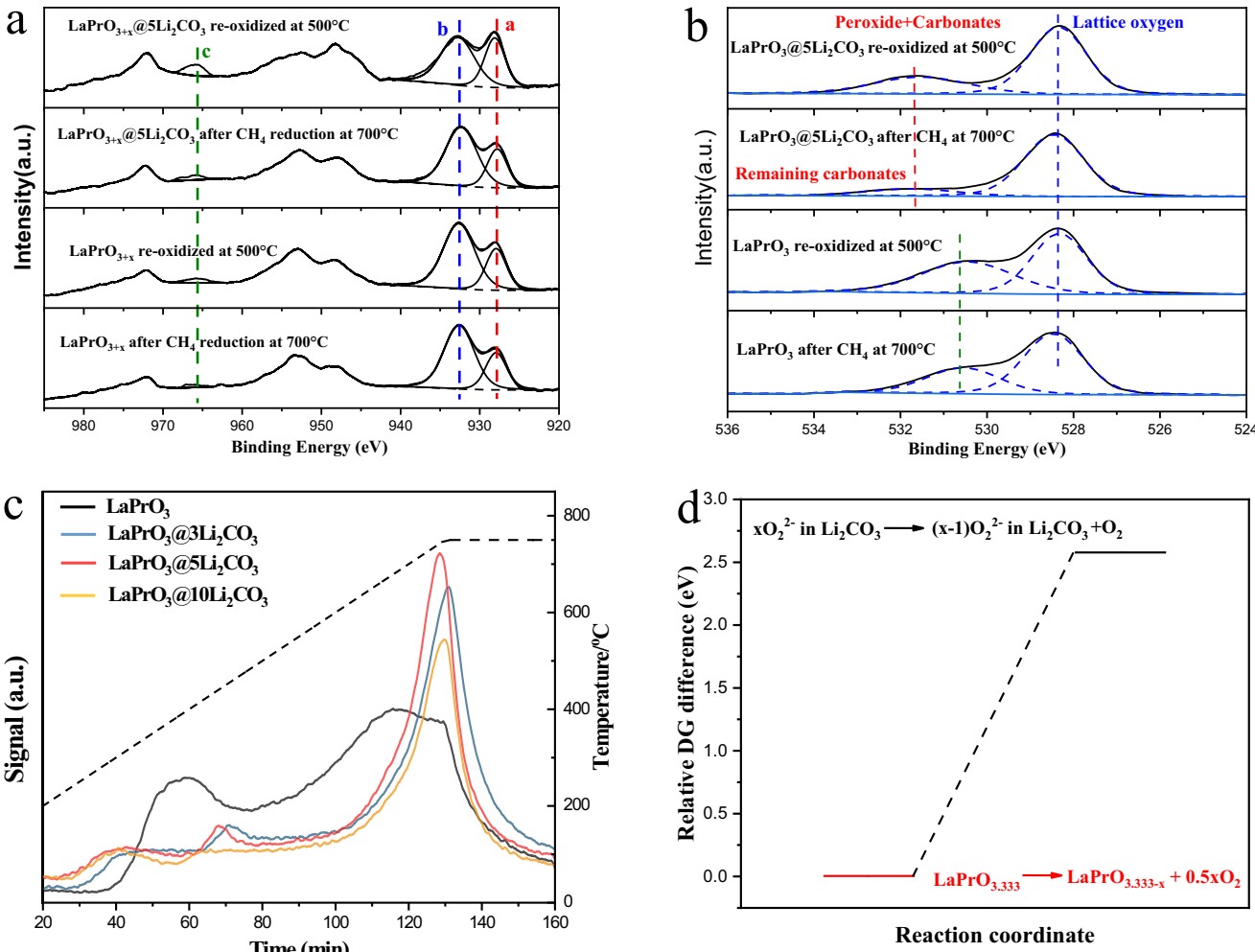

**Fig. 3 | Probe of oxygen species evolution.** In-situ XPS spectra on $LaPrO_{3+x}$ and $LaPrO_{3+x}@5Li_2CO_3$ after methane reduction and oxidation treatments: methane reduction was conducted at 700 °C and re-oxidation was conducted at 500 °C: **a** and **b** show Pr 4$f$ and O 1$s$ peaks, respectively; **c** $O_2$-TPD from $LaPrO_{3+x}$ and $LaPrO_{3+x}@Li_2CO_3$ with different $Li_2CO_3$ loadings; **d** Relative $\Delta G$ difference of $O_2$ release from $LaPrO_{3.33}$ and $Li_2O_2$ in amorphous $Li_2CO_3$.

exhibited much more intense $Pr^{4+}$ features than $LaPrO_{3+x}$ in its oxidized state, confirming the abundance of $Pr^{4+}$ in the near surface region in the presence of the $Li_2CO_3$ coating. It is noteworthy that $Pr^{4+}$ in the near surface region of $LaPrO_{3+x}@5Li_2CO_3$ is largely transformed into $Pr^{3+}$ after contacting methane, as indicated by the decreased "c" peak area and "a/b" peak area ratio. In comparison, the changes in the $Pr^{4+}$ features were quite unremarkable in Li-free $LaPrO_{3+x}$ when exposed to methane. This is likely related to spontaneous decomposition of $LaPrO_{3+x}$ via the reduction of $Pr^{4+}$ in the surface region without the $Li_2CO_3$ layer, under the low oxygen partial pressure in the in-situ XPS (~1 mbar). The corresponding in-situ XPS O 1$s$ spectra are presented in Fig. 3b. $LaPrO_{3+x}$ exhibited an O 1$s$ peak at B.E. = at 528.2 eV that is assigned to lattice oxygen species. The shoulder O 1$s$ peak at B.E. = 530.7 eV for $LaPrO_{3+x}$ has not been reported previously. It is thought to arise from stable hydroxyls to the bare $LaPrO_{3+x}$ since it is independent of the reduction and oxidation treatments. $LaPrO_{3+x}@5Li_2CO_3$ showed two XPS O 1$s$ peaks: B.E. = 528.2 eV, which corresponds to lattice oxygen and does not vary substantially upon reduction or oxidation; and B.E. = 531.8 eV, which is consistent with previous literature assignments of peroxide[37]. This assignment for peroxide is further substantiated by the fact that it decreased substantially when contacting methane, and increased upon re-oxidation with $O_2$. In-situ FTIR-DRIFTS on $LaPrO_{3+x}@5Li_2CO_3$ further confirmed that the carbonate peak increased when methane was injected onto

the sample (Supplementary Fig. S6). The abundance of peroxide species in $LaPrO_{3+x}@5Li_2CO_3$, and their absence in $LaPrO_{3+x}$, can be explained by: (a) the presence of the $Li_2CO_3/Li_2O$ layer, which inhibits peroxide decomposition into molecular $O_2$ from the surface of $LaPrO_{3+x}$ and (b) the increased presence of $Pr^{4+}$ compared to $LaPrO_{3+x}$ as confirmed by XANES and in-situ XPS, which favors $O_2^{2-}$ formation.

The stoichiometric x value in Li-free $LaPrO_{3+x}$ was determined to be 0.33 by thermogravimetric analysis (TGA) upon methane reduction at 700 °C (Supplementary Fig. S7). $O_2$-TPD were further conducted on $LaPrO_{3+x}$ and $LaPrO_{3+x}@5Li_2CO_3$ (Fig. 3c), showing that $LaPrO_{3+x}$ exhibited substantial $O_2$ release at much lower temperatures than $LaPrO_{3+x}@5Li_2CO_3$. Meanwhile, $LaPrO_{3+x}@Li_2CO_3$ with 3–10 wt.% $Li_2CO_3$ loadings all exhibited a primary $O_2$ release peak at ~750 °C. The suppressed peroxide decomposition to gaseous molecular $O_2$ was further examined via ab-initio molecular dynamics (AIMD). As shown in Fig. 3d, $O_2$ formation from $LaPrO_{3+x}$ without the $Li_2CO_3$ shell is far more facile than gaseous molecular $O_2$ formation from $Li_2O_2$ in the amorphous $Li_2CO_3$ thin film. The detailed structural changes of both reactions are shown in Supplementary Fig. S8. This is consistent with the higher oxygen release peak temperature in Fig. 3c and indicates that the $Li_2CO_3$ layer stabilizes the $O_2^{2-}$ peroxide species formed from $LaPrO_{3+x}$. Furthermore, $Li_2CO_3$ has been shown to have a substantial solubility of $O_2^{2-}$ peroxide species[38].

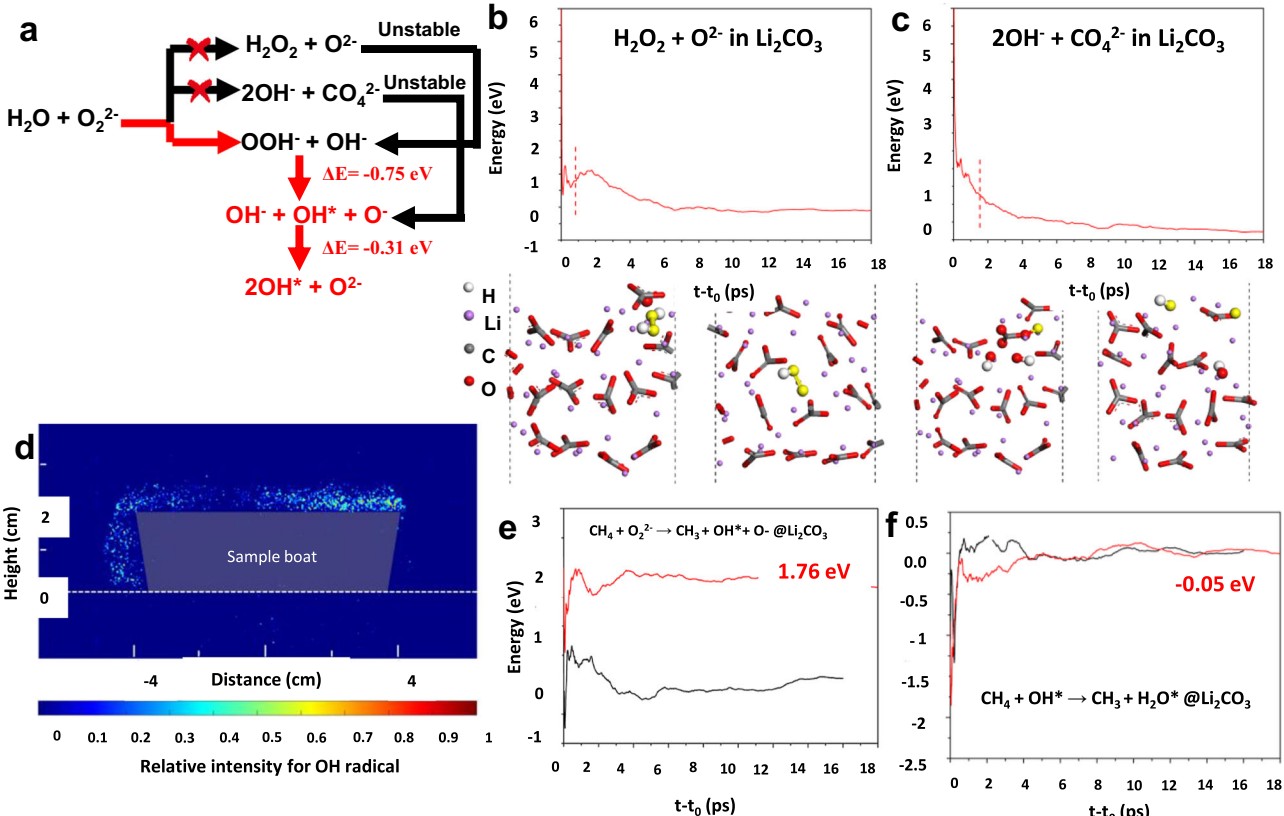

**Fig. 4 | Probe of OH radical evolution. a** Summary of the possible reaction product of $H_2O + O_2^{2-}$; **b, c:** Mean energies as a function of elapsed time ($t$-$t_0$) for evolution of $H_2O_2 + O^{2-}$ and $OH^- + CO_4^{2-}$ in molten $Li_2CO_3$, respectively. The electrophilic oxygen atoms that are involved in the reactions are highlighted in yellow to provide better visualization; **d** LIF experiments on $SiO_2@5Li_2CO_3$, scale bar shows the relative intensity for OH radical; **e** and **f:**, respectively.

## Active redox species and reaction pathway

A detailed AIMD study was conducted to determine the fate of peroxide in $Li_2CO_3$. Interaction between $O_2^{2-}$ and water has been reported to yield OH radicals ($OH^{\cdot}$) in OCM reactions[37]. We further investigated groups of possible products and corresponding reaction pathways using AIMD. As can be seen in Fig. 4a, peroxide is more favorable to evolve into hydroxyl radicals by interacting with $H_2O$ dissolved in the salt. Clearly, $H_2O_2 + O^{2-}$ and $2OH^- + CO_4^{2-}$ cannot be stable because they spontaneously convert to $OOH^- + OH^-$ (Fig. 4b) and $OH^- + OH^{\cdot} + O^-$ (Fig. 4c), respectively. It was calculated that $OH^- + OH^{\cdot} + O^-$ can further evolve into $2OH^{\cdot} + O^{2-}$, intensifying $OH^{\cdot}$ production. This was confirmed with LIF experiments on $Li_2CO_3$ coated $SiO_2$ as a model material at 700°C under $O_2$ and steam, which can detect the formation of $OH^{\cdot}$ (Fig. 4d). The interactions between active species and methane were further studied. It was demonstrated that direct C-H bond activation in methane by $O_2^{2-}$ is not energetically favorable (Fig. 4e). In comparison, the as-formed hydroxyl radical is highly active for methane activation (Fig. 4f). This is also consistent with previous literature report on $Mn-Na_2WO_4/SiO_2$ catalysts[44]. Based on the abovementioned experimental and simulation results, the reaction pathway involves peroxide formation on the $LaPrO_{3+x}$ surface resulting from $Pr^{4+} \rightarrow Pr^{3+}$ transition, dissolution of the $O_2^{2-}$ in the carbonate phase, and subsequent hydroxyl radical formation and $CH_3$ radical formation by C-H bond cleavage. The surface initiated radical reaction will further drive $C_{2+}$ formation in the gas phase[45].

## Catalyst reactivity performance for OCM

The Li-free $LaPrO_{3+x}$ exhibited 57.6% methane conversion, but only 5.14% $C_{2+}$ selectivity with $CO_2$ as the main product at 700°C and 1050 hr$^{-1}$ gas hourly space velocity (GHSV). $Li_2CO_3$ promotion significantly increases the $C_{2+}$ selectivity of $LaPrO_{3+x}$. Figure 5 summarizes

the effects of reaction temperature, space velocity, and methane partial pressure for $LaPrO_{3+x}@5Li_2CO_3$. Higher temperature led to higher methane conversion and $C_{2+}$ yield, but with increased $CO_2$ selectivity (Fig. 5a). At 700 °C under pure methane ($P_{CH4} = 1$ atm), decreasing GHSV led to increased methane conversions, with only a slight decrease in $C_{2+}$ selectivity (Fig. 5b). A maximum $C_{2+}$ yield of 30.6% was obtained at 180 h$^{-1}$, the lowest GHSV tested due to instrumentation limitations. The effect of the OCM step duration was investigated, with the optimum duration determined to be 60 s under the current reactor setting. Longer OCM steps decreased methane conversion, while shorter steps reduced $C_{2+}$ selectivity (Supplementary Fig. S9). We note that the results at all GHSV investigated exceeded the generally accepted "100% rule" for OCM, namely that the sum of methane conversion and $C_{2+}$ selectivity does not exceed 100%[46]. The ability to use undiluted methane represents another advantage from a practical standpoint, when compared to most of the previous literature studies that employed significant amounts of diluent. The effect of methane partial pressure was further illustrated in Supplementary Fig. S10. As can be seen, the space-time yield for ethane, ethylene and $CO_2$ increased almost linearly with increased methane partial pressure from 0.2 atm to 1.5 atm. This suggests a first-order kinetics for both $C_2$ and $CO_x$ formation. Therefore, the $LaPrO_{3+x}@5Li_2CO_3$ redox catalyst can operate at elevated methane partial pressures, which would be highly beneficial for downstream separation and processing of the $C_{2+}$ products. We also note that many of the previously reported OCM catalysts suffered from severe yield penalty at elevated methane partial pressures[26,47,48]. These findings highlight the advantages of $LaPrO_{3+x}@5Li_2CO_3$ in CL-OCM. Figure 5c compares $LaPrO_{3+x}@5Li_2CO_3$ with previously reported OCM catalysts[49–60]: $LaPrO_{3+x}@5Li_2CO_3$ exhibited the highest OCM yield reported so far, and is the only catalyst that exceeds the 30% single-pass $C_{2+}$ yield at 100% methane partial

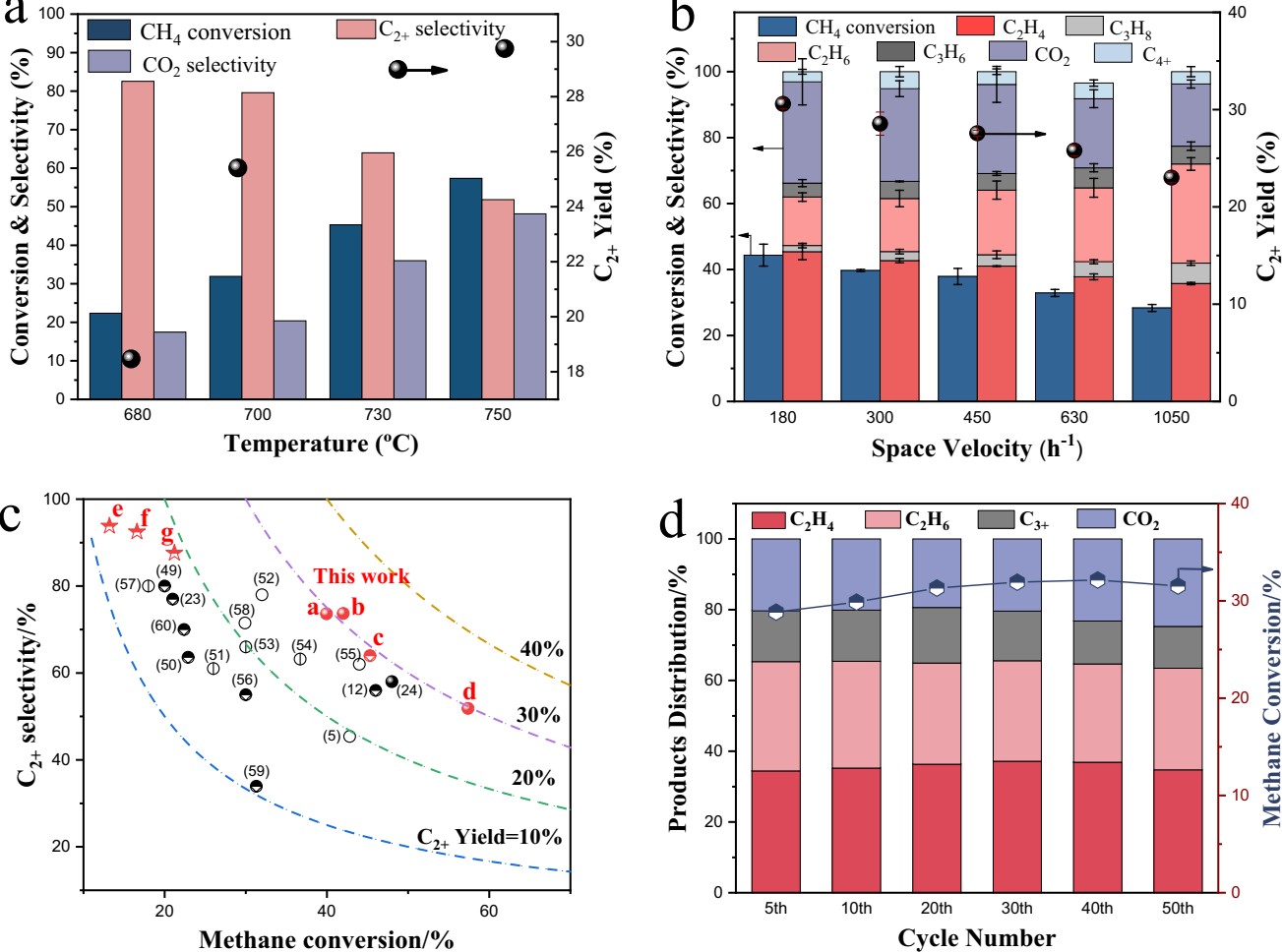

**Fig. 5 | Li₂CO₃ coated LaPrO₃₊ₓ as a redox catalyst for CL-OCM.** Catalyst performance for CL-OCM: **a** Temperature effect for LaPrO₃₊ₓ@5Li₂CO₃: $P_{CH_4}$ = 0.4, GHSV = 1050 h⁻¹; **b** Space velocity effect for LaPrO₃₊ₓ @5Li₂CO₃: T = 700 °C, $P_{CH_4}$ = 1.0. The error bars were expressed as the standard deviations of triplicate experiments. **c** Comparison of LaPrO₃₊ₓ@5Li₂CO₃ with previously reported OCM catalysts: full-filled, half-filled and empty dots represent $P_{CH_4}$ = 0.8–1, 0.4–0.8 and <0.4 atm, respectively. Seven data points from this work are included: **a** T = 700 °C,

$P_{CH_4}$ = 1.0 atm, GHSV = 180 h⁻¹; **b** T = 700 °C, $P_{CH_4}$ = 1.0 atm, GHSV = 300 h⁻¹; **c** T = 730 °C, $P_{CH_4}$ = 0.4 atm, GHSV = 1050 h⁻¹; **d** T = 750 °C, $P_{CH_4}$ = 1.0 atm, GHSV = 180 h⁻¹; **e**–**g** stands for a Mn-Na₂WO₄/SiO₂ catalyst tested under a redox mode at $P_{CH_4}$ = 0.4 atm and GHSV = 1050 h⁻¹, with reaction temperature = 700, 750 and 800° respectively; **d** 50-redox cycle test on LaPrO₃₊ₓ@5Li₂CO₃: T = 700 °C, $P_{CH_4}$ = 0.4, GHSV = 1050 h⁻¹.

pressure. The optimal operating temperature at 700 °C is also significantly lower than the classical OCM catalysts such as Mn-Na₂WO₄/SiO₂, which exhibits optimal performance at ~850 °C. Traditional Mn-Na₂WO₄/SiO₂ catalyst was also synthesized and tested under redox OCM to compare with LaPrO₃₊ₓ@5Li₂CO₃. C₂₊ yields of 12.4%, 15.4% and 18.6% were observed at 700, 750 and 800 °C respectively, indicating that LaPrO₃₊ₓ@5Li₂CO₃ is superior especially at lower reaction temperatures. The LaPrO₃₊ₓ@5Li₂CO₃ catalyst exhibited excellent catalyst stability, as confirmed by long-term performance tests at 700 °C and 1050 h⁻¹ GHSV. Both methane conversion and C₂₊ selectivity were stable within 50 redox cycles as shown in Fig. 5d. In comparison, Li/MgO tends to deactivate after contacting methane and O₂ at 750 °C[34]. The high stability of LaPrO₃₊ₓ@5Li₂CO₃ is ascribed to the preservation of the amorphous Li₂CO₃ overlayer. This was confirmed via ex-situ XPS. As can be seen in Supplementary Fig. S11, the carbonate O 1s peak portion for LaPrO₃₊ₓ@5Li₂CO₃ does not decrease after redox cycles, indicating that surface Li₂CO₃ is maintained. This was also separately validated via a TGA based cyclic experiment (Supplementary Fig. S12). This is substantially different from literature reports on Li/MgO, where Li content decreased from 3.1 wt.% to ~0.1 wt.% within 20 h[61]. The preservation of Li in LaPrO₃₊ₓ@5Li₂CO₃ is likely due to the

lower reaction temperature and the abundance of Li₂CO₃ relative to LiOH, thereby inhibiting Li evaporation[10]. Carbon deposition was negligible after the long-term cycle, as proven by the absence of CO and CO₂ during the re-oxidation step (Supplementary Fig. S13). We also note that under the reaction temperature, Li₂CO₃ could partially decompose into Li₂O, while the as-formed Li₂O can react with the by-product CO₂ in the OCM step and re-form Li₂CO₃. Thus, the catalyst surface is likely to be in a mixed state of Li₂CO₃ and Li₂O at any reaction stage. This is proven by using LiNO₃ instead of Li₂CO₃ for wet impregnation onto LaPrO₃₊ₓ, while keeping the same Li amount. The as-synthesized LaPrO₃₊ₓ@Li₂O (after nitrate decomposition) started to exhibit activity for OCM after a few reaction cycles, although the C₂₊ yield is lower than that of LaPrO₃₊ₓ@Li₂CO₃ (Supplementary Fig. S14). The presence of Li₂O in Li₂CO₃ can be beneficial for the formation of Li₂O₂ by reacting with the active oxygen species on the LaPrO₃₊ₓ surface.

## Generalizability of the OCM catalyst design strategy

The core-shell redox catalyst design strategy can be extended to other Pr-containing mixed lanthanide oxides. NdPrO₃₊ₓ, DyPrO₃₊ₓ, SmPrO₃₊ₓ, HoPrO₃₊ₓ and EuPrO₃₊ₓ were synthesized using a similar

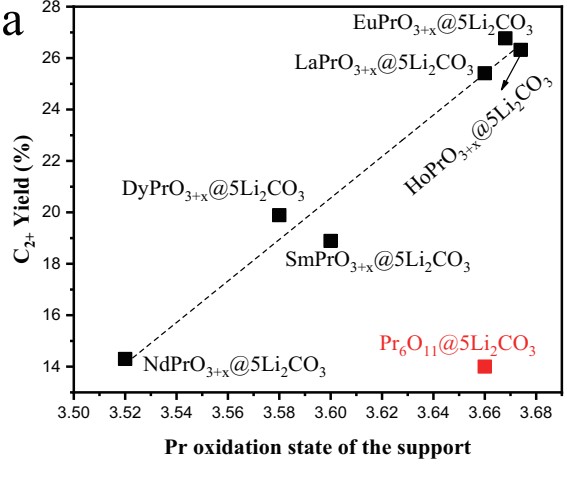

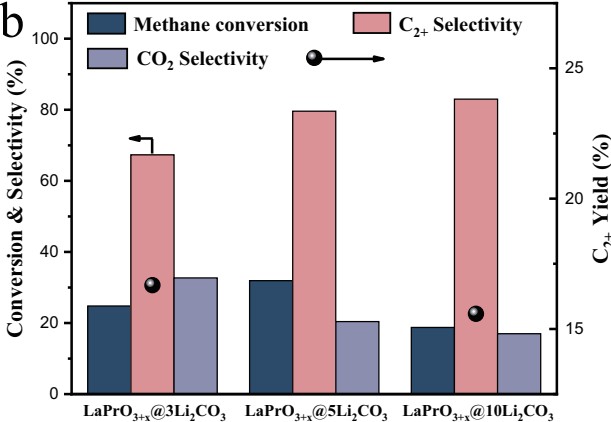

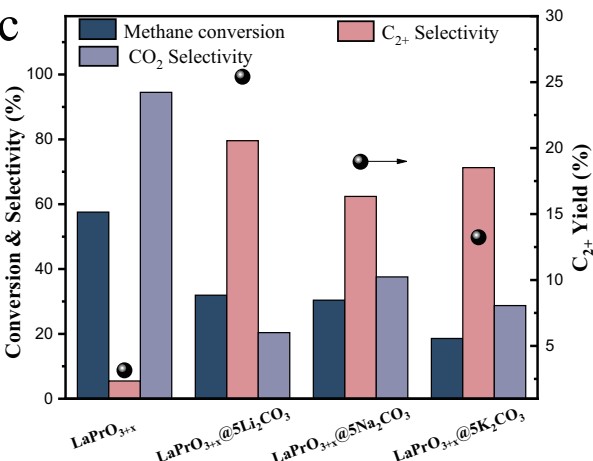

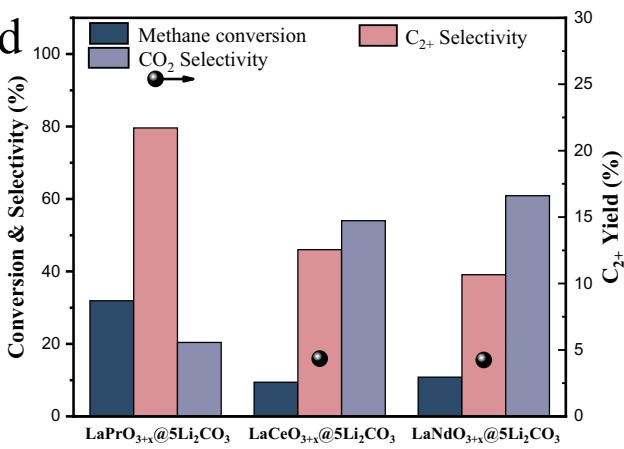

**Fig. 6 | Extension of the LaPrO$_{3+x}$@5Li$_2$CO$_3$ redox catalyst. a** Plots of C$_{2+}$ yields with Pr oxidation states of different mixed lanthanide oxide supports; **b** Redox OCM performance comparison of LaPrO$_{3+x}$@3Li$_2$CO$_3$, LaPrO$_{3+x}$@5Li$_2$CO$_3$ and LaPrO$_{3+x}$@10Li$_2$CO$_3$; **c** Redox OCM performance comparison of LaPrO$_{3+x}$, LaPrO$_{3+x}$@Li$_2$CO$_3$, LaPrO$_{3+x}$@5Na$_2$CO$_3$ and LaPrO$_{3+x}$@5K$_2$CO$_3$; **d** Redox OCM performance comparison of LaPrO$_{3+x}$@5Li$_2$CO$_3$, LaCeO$_{3+x}$@5Li$_2$CO$_3$ and LaNdO$_{3+x}$@5Li$_2$CO$_3$. Reaction conditions for **b**–**d**: $T$ = 700 °C, $P_{CH4}$ = 0.4 atm, $GHSV$ = 1050 h$^{-1}$.

method. The Pr oxidation states of the carbonate-free mixed oxides were determined by TGA measurement of oxygen stoichiometry upon methane reduction at 700 °C. The mixed oxides were then loaded with 5 wt.% Li$_2$CO$_3$ and examined for OCM. The C$_{2+}$ yields, plotted in Fig. 6a, correspond to the Pr oxidation state in the mixed oxides. A linear relationship is observed between the C$_{2+}$ yield and the initial Pr oxidation state. EuPrO$_{3+x}$@5Li$_2$CO$_3$ and HoPrO$_{3+x}$@5Li$_2$CO$_3$ even achieved slightly better C$_{2+}$ yields than LaPrO$_{3+x}$@5Li$_2$CO$_3$. The significantly reduced C$_{2+}$ yield of Pr$_6$O$_{11}$@5Li$_2$CO$_3$, despite the Pr oxidation state in Pr$_6$O$_{11}$ being +3.67, can be attributed to the instability of the Pr$_6$O$_{11}$ phase in the presence of Li$_2$CO$_3$. This leads to the formation of the Li$_{26}$Pr$_{36}$O$_{73}$ phase after cycling. (see Supplementary Fig. S15a) and consistent with the report by Aono et al., who observed the same phase by heating up the a Pr$_6$O$_{11}$ and Li$_2$CO$_3$ mixture[62]. The decrease in the Pr oxidation state and disruption of the Li$_2$CO$_3$ layer renders low C$_{2+}$ yield. In comparison, all the Li$_2$CO$_3$ promoted mixed lanthanide oxides, which the exception of NdPrO$_{3+x}$, maintained their original phases after cycling, and no Li-containing phases were observed (Supplementary Figs. S15b−e). This also highlights the importance of the secondary rare earth metal cation such as La and Sm, which stabilizes Pr$^{4+}$ and inhibits the solid-state reaction with the Li$_2$CO$_3$ promoter. The Li$_2$CO$_3$ loading effect on LaPrO$_{3+x}$ was also investigated as shown in Fig. 6b. Although the oxygen release behaviors were all substantially altered with different Li$_2$CO$_3$ loadings as shown in O$_2$-TPD

(Fig. 3c), the loading amount exhibits an optimum and LaPrO$_{3+x}$@5Li$_2$CO$_3$ achieved the highest C$_{2+}$ yield. In addition to Li$_2$CO$_3$, other alkali metal carbonate promoters including Na$_2$CO$_3$ and K$_2$CO$_3$ were also investigated as shown in Fig. 6c. Switching from Li$_2$CO$_3$ to Na$_2$CO$_3$ and K$_2$CO$_3$ leads to decreased catalyst activities and decreased C$_{2+}$ yields. This may be due to the lower activities of Na$_2$O$_2$ and K$_2$O$_2$, where Na$_2$O$_2$ and K$_2$O$_2$ are more thermodynamically stable than Li$_2$O$_2$ (Supplementary Table S2). The necessity of Pr in the mixed metal oxide support was also investigated by switching Pr to Ce and Nd, where Ce and Nd locating very close to Pr in the periodic table in the lanthanide family. As shown in Fig. 6d, both LaCeO$_{3+x}$@5Li$_2$CO$_3$ and LaNdO$_{3+x}$@5Li$_2$CO$_3$ exhibited very low C$_{2+}$ yields and high selectivities towards CO$_2$. This is probably due to the properties of the Pr$^{4+}$↔Pr$^{3+}$ redox pair, which leads to efficient generation of peroxide oxygen species in the Li$_2$CO$_3$ salt.

## Discussion

In this work, we present a generalized strategy for the chemical looping - oxidative coupling of methane (CL-OCM) using Li$_2$CO$_3$-promoted mixed rare earth oxides. A detailed study on LaPrO$_{3+x}$@Li$_2$CO$_3$ revealed a single-pass C$_{2+}$ yield of up to 30.6% with good catalyst stability at 700°C. Additionally, the operational partial pressure of methane can exceed 1 atm, offering potential industrial benefits. The Li$_2$CO$_3$ promotion formed a surface layer on LaPrO$_{3+x}$, increasing both

the bulk and surface $Pr^{4+}$ contents, which in turn enhanced OCM activity. This finding was corroborated by ex-situ XANES, in-situ Raman, in-situ XRD, and in-situ XPS analyses. In-situ Raman and XPS measurements also suggested that $Pr^{4+}$ contributed to the presence of near surface peroxide on $LaPrO_{3+x}@Li_2CO_3$. The peroxide species would subsequently transform into hydroxyl radicals for methane activation, as supported by both LIF experiments and AIMD simulations. A generalized correlation between the oxidation state of Pr in the mixed lanthanide oxide and $C_{2+}$ yield was also observed, providing a valuable strategy for optimizing this family of OCM catalysts. Given their high yields and favorable operational parameters, $Li_2CO_3$-promoted mixed rare earth oxides hold great promise for the direct conversion of methane to $C_{2+}$ products.

## Methods

### Redox catalyst preparation

A modified Pechini method was used to prepare $LaPrO_{3+x}$. Stoichiometric amounts of $La(NO_3)_9 \cdot 6H_2O$ (99.0%, Sigma-Aldrich, 10 g) and $Pr(NO_3)_9 \cdot 6H_2O$ (99.0%, Sigma-Aldrich, 10 g) were dissolved in 100 ml deionized water and stirred to form a transparent solution. Citric acid (99.5%, Sigma-Aldrich, 28 g) at a 3:1 molar ratio to metal ions, and ethylene glycol (99.8%, Sigma-Aldrich, 18 ml) at a 2:1 molar ratio to citric acid were added into the solution. The obtained solution was stirred constantly at 80 °C to form a viscous gel. After that, the gel was transferred to a convection oven for drying at 130 °C overnight and was then calcined in a tube furnace at 850 °C for 8 h. A wet impregnation method was used to synthesize $LaPrO_{3+x}@Li_2CO_3$. Calculated amount of $Li_2CO_3$ (ACS reagent; ≥99.0%) was dissolved in 10 ml deionized water. 5 g of $LaPrO_{3+x}$ was added into the solution under stirring. After drying at 130 °C for 2 h, the dried particles were calcined in a furnace at 750 °C for 3 h. Finally, the powders were ground, pressed and crushed into 60–80 mesh as final $LaPrO_{3+x}@Li_2CO_3$.

### Redox catalyst characterization

Redox catalyst surface and morphology characterizations were conducted with ex-situ and in-situ XRD, ex-situ and in-situ XPS, ex-situ and in-situ S/TEM, in-situ Raman, quasi in-situ LEIS, in-situ DRIFTS-FTIR and ex-situ XANES. Ex-situ XRD was conducted with a Rigaku SmartLab X-ray diffractometer at NC State University. In-situ XRD was conducted on an Empyrean X-ray diffractometer equipped with an Anton-Paar XRK-900 reactor chamber at NC State University. Ex-situ XPS was conducted on an ESCALAB 250Xi (Thermo Fisher) at Guangzhou Institute of Energy Conversion. In-situ XPS was conducted with SPECS EnviroESCA at Dalian Institute of Chemical Physics. S/TEM were conducted on an aberration corrected Thermo Scientific Titan 80-300 STEM at NC State University. In-situ Raman was conducted on a Horiba LabRam-HR Raman spectrometer at Lehigh University. Quasi in-situ HS-LEIS was conducted at the Surface Analysis Center at Lehigh University with an ION-TOF Qtac[100] for outermost surface layer compositional analysis and depth profiling. In-situ DRIFTS-FTIR was conducted on a Thermo Fisher Nicolet iS50 FTIR equipped with a DiffusIR sample chamber (Pike Technologies) at NC State University. Ex-situ XANES was conducted on an X-ray Absorption Fine structure for catalysis (XAFCA) with an ion-chamber detector at Singapore Synchrotron Light Source. Characterization details of each method can be found in the supplemental document.

### Reactivity tests

Reactivity tests were conducted in a fixed bed quartz U-tube reactor with ID of 1/8 inches or 3.18 mm. Approximately 2 g of catalyst was loaded at the bottom of the U-tube reactor with quartz wool placed on both sides of the reactor to keep the catalysts in place. Typically, the OCM reaction was conducted at 700 °C, a mixture of methane

(20–100%, balance Ar) was injected into the reactor for 1 min. After the OCM step, Ar was introduced to purge the reactor for 5 min and then 10% oxygen (5 mL/min, balance Ar) was introduced for the oxidation step for 3 min. A gas bag was used to collect all the gas product over the entire OCM step. The obtained gaseous products collected were detected by gas chromatography (Agilent 7890 A). To confirm the redox stability of the redox catalyst, 50 reduction and oxidation steps were performed following the above procedure, with 5 min of Ar purge in between. The catalyst OCM activity are calculated based on the average products across the OCM step obtained in the gas bag. The equations used for calculating conversions, selectivities and yields are:

$$\text{Methane Conversion} = \frac{\text{Methane Input} - \text{Methane Output}}{\text{Methane Input}} \quad (1)$$

$$\text{C2+ Selectivity} = \frac{\text{moles of C in C2+ products}}{\text{moles of C in converted methane}} \quad (2)$$

$$\text{C2+ Yield} = \text{Methane Conversion} * \text{Selectivity of C2+} \quad (3)$$

### Computational details

AIMD calculations were implemented by the Vienna ab initio Simulation package (VASP) with the frozen-core all-electron projector augmented wave (PAW) model and Perdew-Burke-Ernzerhof (PBE) functions. A kinetic energy cutoff of 350 eV is used for the plane-wave expansion of the electronic wave function, and a Γ-point is chosen for sampling the first Brillouin zone. The convergence criteria of force and energy were set to 0.01 eV/Å and $10^{-5}$ eV respectively. The strong on-site coulomb interaction on the d-orbital electrons on the Fe sites is treated with the generalized gradient approximation (GGA) + U approach with $U_{eff} = 4$ eV for the f-orbital of Pr. Spin polarization is included in all calculations. Constant temperature AIMD simulations are performed at 1000 K, which is slightly above the experimental reaction temperature (700 °C). The atomic motions are treated classically and propagated with 1 fs time steps.

The internal energy of all molten systems is obtained from the AIMD simulations as the time average kinetic and potential energy: $E(t) = \frac{1}{t-t_0} \int_{t_0}^{t} (E_{DFT}(\tau) + E_{kin}(\tau)) d\tau$[6], where $t_0$ is chosen to allow the system to equilibrate and lose memory of the initial conditions, which was set as 10 ps unless otherwise stated. For the gas-molecules, $E(t)$ are corrected by adding the translational energy $\frac{3}{2}k_BT$ because it contains only rotational and vibrational contributions, where $k_B$ is the Boltzmann constant. The estimated change in Gibbs free energy is obtained as $\Delta G_{estimate}^{\circ} = \Delta E + p\Delta V - T\Delta S_{estimate}^{\circ}$, where the volume change ($\Delta V$) is assumed to originate purely from changes in the number of gas phase molecules ($\Delta n_{gas}$) and is calculated by the ideal gas law ($p\Delta V = \Delta n_{gas}k_BT$). The entropies of the studied radicals are obtained from NIST.

## Data availability

The source data generated in this study are provided in the Source Data file and are also available from the corresponding author upon reasonable request. All other data are available from the corresponding author upon request. All data needed to evaluate the conclusions in the paper are present in the paper and/or the Supplementary Materials (including Supplementary Figs. 1–15, details of the instrumentation, additional XRD, XPS, Raman and thermogravimetric analysis). The source data for the figures are all provided with this paper. Source data are provided with this paper.

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

## Acknowledgements

This work was supported by the U.S. NSF (Award No. CBET-2116724, CBET-1923468) and the Kenan Institute for Engineering, Technology and Science at the NC State University. KZ is supported by Guangdong Natural Science Fund for Distinguished Young Scholars (Grants 2023B1515020048). We also acknowledge the support from the Deutsche Forschungsgemeinschaft (DFG, German Research Foundation) under Germany's Excellence Strategy – EXC 2008/1 (UniSysCat) – 390540038 and the Alexander von Humboldt Foundation. We acknowledge the use of the Analytical Instrumentation Facility (AIF) at the NC State University.

## Author contributions

F.L. conceived and supervised the study. K.Z. and Y.G. conducted the experimental work and characterizations. Y.G. coordinated the characterizations and data interpretation. K.Z., Y.G. and F.L. wrote the manuscript. I.W. supervised the in-situ Raman characterizations and edited the manuscript. B.M. performed in-situ Raman and XPS characterizations. X.W. (Xinjun Wang) conducted the AIMD calculations. J.L., B.J. and X.W. (Xin Wang) conducted part of the experimental work. C.H., W.G. and X.W. (Xiaodong Wang) performed the in-situ XPS characterizations. A.Z. and Z.H. supervised the characterizations. J.S., J.H. and R.S. edited the manuscript.

## Competing interests

The authors declare no competing interests.
