## [Peer Review File · Nature Communications]

Lithium carbonate-promoted mixed rare earth oxides as a generalized strategy for oxidative coupling of methane with exceptional yieldsREVIEWER COMMENTS

Reviewer #1 (Remarks to the Author):

Comments:

The authors present a comprehensive study on the use of Li_2CO_3 -coated mixed rare earth oxides as redox catalysts for the oxidative coupling of methane (OCM) under a chemical looping scheme. They investigated a series of Pr-containing lanthanide oxides with a Li_2CO_3 promoter, denoted as $\text{LnPrO}_{3+x}@\text{Li}_2\text{CO}_3$ ($\text{Ln} = \text{La}, \text{Eu}, \text{Ho}, \text{Dy}, \text{Sm}, \text{and Nd}$) for C_2^+ production. The results demonstrate that this family of materials achieved a single-pass C_2^+ yield of up to 30.6% at 700°C . They also revealed that the Li_2CO_3 coating on LaPrO_{3+x} played a significant role, leading to an increase in both bulk and surface Pr^{4+} contents, thus enhancing OCM activity. The findings are important to researchers in the field. I recommend it be published after the following comments are addressed:

The authors conducted in-situ XRD, Raman, and XPS measurements to monitor the dynamics of the bulk LaPrO_{3+x} phase for both LaPrO_{3+x} and $\text{LaPrO}_{3+x}@\text{Li}_2\text{CO}_3$ under oxidizing and methane reducing conditions. The presence of peroxide species was observed in $\text{LaPrO}_{3+x}@\text{Li}_2\text{CO}_3$, whereas they were absent in the Li-free LaPrO_{3+x} sample. Based on these findings they suggested that the peroxide species are associated with the thin Li_2CO_3 shell. The authors should clearly explain how Li_2CO_3 affects the formation of peroxide species. They should also discuss the impact of the thickness of the shell on the dynamics of the bulk LaPrO_{3+x} phase.

Given the relatively low surface area of $\text{LaPrO}_{3+x}@\text{Li}_2\text{CO}_3$ ($\sim 1 \text{ m}^2/\text{g}$), the authors concluded that the detected peroxide signal by Raman could be attributed to surface-bound species and bulk peroxide species. To support this conclusion, the authors should provide the surface area values of other samples, such as LaPrO_3 , $\text{LaPrO}_3@3\text{Li}_2\text{CO}_3$, and $\text{LaPrO}_3@10\text{Li}_2\text{CO}_3$, and compare the peroxide signals of these different samples. This analysis will strengthen the authors' conclusion and provide insights into the influence of surface area on the presence of peroxide species in the studied materials. In addition, Raman did not detect Li_2CO_3 peaks from the $\text{LaPrO}_{3+x}@\text{Li}_2\text{CO}_3$ sample. It is unclear if it is due to the low loading. To clarify the loading effect, the authors should also examine the Li_2CO_3 peaks of $\text{LaPrO}_3@3\text{Li}_2\text{CO}_3$ and $\text{LaPrO}_3@10\text{Li}_2\text{CO}_3$ samples.

Fig. 5c presents a comparison of $\text{LaPrO}_{3+x}@\text{Li}_2\text{CO}_3$ with previously reported OCM catalysts. The authors claimed that $\text{LaPrO}_{3+x}@\text{Li}_2\text{CO}_3$ exhibits the highest OCM yield reported to date and is the only catalyst surpassing the 30% single-pass C_2^+ yield at 100% methane partial pressure. Additionally, they highlighted that the optimal operating temperature for $\text{LaPrO}_{3+x}@\text{Li}_2\text{CO}_3$ (700°C) is lower than that of classical OCM catalysts like $\text{Mn-Na}_2\text{WO}_4/\text{SiO}_2$ ($\sim 850^\circ\text{C}$). However, it may not be appropriate to directly compare the performance of $\text{LaPrO}_{3+x}@\text{Li}_2\text{CO}_3$ in a CL-OCM experiment to that of a $\text{Mn-Na}_2\text{WO}_4/\text{SiO}_2$ catalyst in a steady-state experiment. The authors should test their material in steady state experiments to compare with the $\text{Mn-Na}_2\text{WO}_4/\text{SiO}_2$ catalyst or alternatively, test the $\text{Mn-Na}_2\text{WO}_4/\text{SiO}_2$ catalyst in a CL-OCM experiment.

The authors found that the hydroxyl radical is highly active for methane activation. However, the

authors didn't further discuss the impact of hydroxyl radical on CH₃ dimerization or decomposition on LaPrO_{3+x}@5Li₂CO₃, which could significantly influence the yield of C₂₊ hydrocarbons.

The authors performed AIMD calculations using the Vienna ab initio Simulation package (VASP). However, they did not include details regarding the convergence criteria used for electronic minimization.

Figure 4(b-f) is difficult to read due to color combinations and small font size. In addition, the authors should clarify what the colored balls in the molecular model represent.

Reviewer #2 (Remarks to the Author):

In this draft, the authors prepared Li₂CO₃-coated mixed rare earth oxides for OCM reaction. It was found that on LaPrO_{3+x}@5Li₂CO₃, one of the three best catalysts, the one-way C₂₊ yield of 30.6% can be achieved at 700 degree C with a chemical loop reaction strategy. The single pass yield of 30% is usually regarded as the minimum requirement to industrialize this interesting and important reaction. Up to date, on all the tried catalysts with a constant CH₄+O₂ feed, it has not accomplished this goal yet. From this point of view, I believe the results reported in this draft is interesting for fundamental study, though the GHSV to obtain this value is very low. With the combination of experimental and DFT calculation, the authors have analyzed the structures of the catalysts in detail and tried to figure out the factors leading to the high yield. They found that the surface layer of Li₂CO₃ is important to maintain the reaction performance of the catalysts, which can stabilize surface peroxide anions and Pr⁴⁺ cations, both are critical to OCM reaction on the catalysts prepared in this study.

The draft can be accepted after addressing the following issues:

1. It is well-known that Mn-Na₂WO₄/SiO₂ is the best catalyst formulation up to date. Did the authors compare its reaction performance with that of LaPrO_{3+x}@5Li₂CO₃ (or one of the best three catalysts prepared by the authors) under the same condition? Either using the chemical loop strategy or traditional fixed-bed set-up with constant CH₄+O₂ flow in lean O₂. Although the authors do a comparison in Figure 5(c), I believe that most of the reaction data were not obtained with a chemical loop method if they were reported in former publications. Thus, I suggest the authors test the reaction performance of Mn-Na₂WO₄/SiO₂ under the same condition for more clear information.
2. As to the title, it could be better to change into "Lithium carbonate-promoted...". The current title will obviously lead people's thought into the formation of active surface carbonate intermediate. But this is not main point. Moreover, 5% is not a low loading for Li₂CO₃, but the authors did not detect it with Raman. Since Raman is very sensitive to micro-crystallites, and the surface area of the sample is very low, this is unusual. In Figure S4, the authors in fact see carbonates with In-situ DRIFTS even in 10% O₂, can this carbonate peak be assigned to Li₂CO₃? If so, then how to explain the increase of carbonate peak after introducing CH₄. As I understand, part of Li₂O is still present in the shell layer.
3. During the chemical loop process, in the step of CH₄ injection, is there any carbon deposits formed. Though the authors have demonstrated that during 50 cycles, the catalyst is regenerable by O₂ and stable, some coking being detrimental to the catalyst structure could still occur after long-term running.

It will be interesting to clarify this issue.

4. Some typos: line 187, "Fig. 4a" should be "Fig. 3a", line 218, "Fig. 4c" should be "Fig. 3c".

Reviewer #3 (Remarks to the Author):

This an interesting and insightful paper, discussing the reaction mechanism of chemical looping experiments for oxidative coupling of methane, using new LaPrO₃-based catalysts. Overall, the results and the conclusions from the results are very exciting. My only comments are about details in the study and the role of Li₂CO₃ vs Li₂O₂. Please see below.

Abstract

Can the authors give specific information about the "practical methane partial pressures"?

Does a catalyst system mean a catalyst composite? Or a novel catalytic scheme for the process?

How does a 30.6% yield compare to previous studies and industrially competitive processes?

Is Pr essential in the catalytic composite – if yes, why not describe it as the best catalyst among those analyzed? Then the section about the Pr oxidation state will be better connected to the rest of the abstract.

What is this optimization strategy (composition, structure, operability state)? Can you give a short description?

Introduction

Please remove and/or in the first paragraph. Again, I object to the use of the word "system" when describing multi-component catalysts. Please, reconsider for this manuscript.

Previous work listed – please clarify whether the experiments were conducted continuously.

Green et al. and Labinger et al.'s modelling studies – please reveal why the C₂+ yield cannot exceed 30-35%.

The authors say: "As such, a gap clearly exists between reported academic research results and industrial application" What are the requirements for industrial application?

"spectroscopy (XPS) spectrometer" something is missing here, perhaps, "using a" spectrometer?

"is isolated, pseudotetrahedral Na-coordinated WO₄ surface sites" – consider correcting to "are isolated pseudotetrahedral Na-coordinated WO₄ surface sites."

Results

"all the Li₂CO₃ promoted LnPrO_{3+x} oxides (Ln = La, Eu, Ho, Dy, Sm, Nd) were active for OCM" – where can we see this?

Fig. 1 b – is EELS used in anything for this result – you only show EDX mapping, or did I misunderstand?

Please confirm whether the EDX (or EELS) image is a cross-section of a particle or an external surface of a particle.

What benefits are of using in-situ TEM in Fig. 1e over ex-situ?

Working with low loading of carbonates is usually associated with problems confirming the presence of the carbonate. The XPS results confirm the presence of Li₂CO₃, but can the authors comment on the phase diagram of this component – is it expected to react or decompose under CH₄ or O₂? Where does the peroxide species come from if it is associated with the carbonate? What happens to carbon during

the creation of Li_2O_2 ? Is the carbonate even needed in cycling, or rather the goal is to have Li_2O_2 (or Li_2O) all along? Have you tried depositing Li_2O and oxidizing it to obtain Li_2O_2 directly?

Fig. 3 – can you identify the oxygen in the carbonate species in O1s for the in-situ XPS (I see it in the SI for ex-situ experiments)?

Is the effect of cooperation between Li_2CO_3 and LaPrO_{3+x} dependent on the extent of the interface or on the molar or mass ratios? Can the authors comment based on their extensive experimental results? I fail to see any meaningful result in Fig. 4 d – the caption says the figure shows the relative intensity for OH radical – in what sense relative? What is on the x, y axes?

Catalyst reactivity – please comment on the longevity of catalysts other than $\text{LaPrO}_3@5\text{Li}_2\text{CO}_3$.

How do the results (mainly product distribution) depend on the time of reduction and oxidation? I do not consider this essential for the paper, as the paper focuses on comparing different materials, but I am curious whether this can be an important factor – so I hope the authors can comment on their decision for the time steps in cycling.

The authors wrote: “In addition to Li_2CO_3 , other alkali metal carbonates such as Na_2CO_3 and K_2CO_3 were also investigated using LaPrO_{3+x} as the redox-active component”. Can you add information on the yields of C_2^+ achieved, or even add that information to Fig. 6?

Methods

Are the results of the catalyst OCM activity from the start of the cycle, the end, or the average across the cycle?

How were selectivity, conversion and yield calculated?

Supplementary

Fig. S6 and Fig. S9 – please add information about the experimental temperature. Can you comment on whether the oxidation step brings the bare LaPrO_{3+x} mass to the original value?

Response to comments from reviewers:

Reviewer 1:

The authors present a comprehensive study on the use of Li_2CO_3 -coated mixed rare earth oxides as redox catalysts for the oxidative coupling of methane (OCM) under a chemical looping scheme. They investigated a series of Pr-containing lanthanide oxides with a Li_2CO_3 promoter, denoted as $\text{LnPrO}_{3+x}@\text{Li}_2\text{CO}_3$ ($\text{Ln} = \text{La}, \text{Eu}, \text{Ho}, \text{Dy}, \text{Sm}, \text{and Nd}$) for C_2^+ production. The results demonstrate that this family of materials achieved a single-pass C_2^+ yield of up to 30.6% at 700°C . They also revealed that the Li_2CO_3 coating on LaPrO_{3+x} played a significant role, leading to an increase in both bulk and surface Pr^{4+} contents, thus enhancing OCM activity. The findings are important to researchers in the field. I recommend it be published after the following comments are addressed:

Response: We thank the reviewer for the positive comments. Our responses to the specific comments are listed below.

1. The authors conducted *in-situ* XRD, Raman, and XPS measurements to monitor the dynamics of the bulk LaPrO_{3+x} phase for both LaPrO_{3+x} and $\text{LaPrO}_{3+x}@\text{Li}_2\text{CO}_3$ under oxidizing and methane reducing conditions. The presence of peroxide species was observed in $\text{LaPrO}_{3+x}@\text{Li}_2\text{CO}_3$, whereas they were absent in the Li-free LaPrO_{3+x} sample. Based on these findings they suggested that the peroxide species are associated with the thin Li_2CO_3 shell. The authors should clearly explain how Li_2CO_3 affects the formation of peroxide species. They should also discuss the impact of the thickness of the shell on the dynamics of the bulk LaPrO_{3+x} phase.

Response: We appreciate the comment. In the revised manuscript, the effect of Li_2CO_3 on the formation of peroxide species was discussed in more detail. Based on the *in-situ* Raman results in Fig. 2 in our manuscript, both LaPrO_{3+x} and $\text{LaPrO}_{3+x}@\text{Li}_2\text{CO}_3$ exhibit Pr^{4+} species at their oxidized state. For $\text{LaPrO}_{3+x}@\text{Li}_2\text{CO}_3$, Pr^{4+} is capable of inducing the formation of Li_2O_2 at the gas-molten salt interface, and Li_2CO_3 is known to have high solubility for peroxide species. This was also confirmed via *in-situ* Raman. Based on AIMD calculations (Fig. 3(d) in the manuscript), we found that the peroxide inside the Li_2CO_3 salt is more difficult to self-decompose into molecular O_2 . In contrast, LaPrO_{3+x} without the Li_2CO_3 shell tends to spontaneously release its lattice oxygen, leading to the deprivation of Pr^{4+} species. This result is also confirmed by *in-situ* XPS (Fig. 3(a) and 3(b) in the manuscript): For instance, $\text{LaPrO}_{3+x}@\text{Li}_2\text{CO}_3$ exhibited more surface Pr^{4+} and peroxide species under 1 mbar partial pressure of O_2 , whereas LaPrO_{3+x} barely exhibited surface Pr^{4+} and peroxide species. We have added these discussions into our manuscript to make it clearer.

The impact of the shell thickness on the dynamics of the bulk LaPrO_{3+x} phase is reflected by the Li_2CO_3 loading effect, as higher Li_2CO_3 loading would lead to a thicker Li_2CO_3 shell. Performance data and O_2 -TPD results suggest that the Li_2CO_3 coverage with different loadings can all effectively modify the LaPrO_{3+x} surface. As shown in Fig. R1a below, 3 wt.% of Li_2CO_3 loading has effectively shifted the oxygen release peaks from LaPrO_{3+x} to much higher temperatures. Moreover, we note that different Li_2CO_3 loadings exhibited similar effects in O_2 -TPD results, showing high-temperature oxygen release peaks at similar positions. This indicates that Li_2CO_3 has similar effects of surface modification in despite of their loadings. However, an optimal loading of Li_2CO_3 indeed exists. Too small of a Li_2CO_3 loading may lead to incomplete surface coverage of the salt, whereas a high Li_2CO_3 loading may lead to lower O_2^{2-} concentration and/or transport limitations for the peroxide oxygen species to the outer layer of the molten salt. Both would lead to decreased C_2^+ yield. This is reflected in Fig R1(b), where we have tested the redox OCM performances of

LaPrO_{3+x}@3Li₂CO₃, LaPrO_{3+x}@5Li₂CO₃ and LaPrO_{3+x}@10Li₂CO₃. As can be seen, LaPrO_{3+x}@5Li₂CO₃ achieved the highest C₂₊ yield.

Figure R1. (a) O₂-TPD from LaPrO_{3+x} and LaPrO_{3+x}@Li₂CO₃ with different Li₂CO₃ loadings; (b) Redox OCM performance comparison for LaPrO_{3+x}@3Li₂CO₃, LaPrO_{3+x}@5Li₂CO₃ and LaPrO_{3+x}@10Li₂CO₃

2. Given the relatively low surface area of LaPrO_{3+x}@5Li₂CO₃ (~1 m²/g), the authors concluded that the detected peroxide signal by Raman could be attributed to surface-bound species and bulk peroxide species. To support this conclusion, the authors should provide the surface area values of other samples, such as LaPrO₃, LaPrO₃@3Li₂CO₃, and LaPrO₃@10Li₂CO₃, and compare the peroxide signals of these different samples. This analysis will strengthen the authors' conclusion and provide insights into the influence of surface area on the presence of peroxide species in the studied materials. In addition, Raman did not detect Li₂CO₃ peaks from the LaPrO_{3+x}@5Li₂CO₃ sample. It is unclear if it is due to the low loading. To clarify the loading effect, the authors should also examine the Li₂CO₃ peaks of LaPrO₃@3Li₂CO₃ and LaPrO₃@10Li₂CO₃ samples.

Response: We thank the reviewer for the suggestion. We have added the surface areas of LaPrO₃, LaPrO_{3+x}@3Li₂CO₃, LaPrO_{3+x}@5Li₂CO₃ and LaPrO_{3+x}@10Li₂CO₃ tested by BET in Table R1. As can be seen, these surface areas are all within the same order of magnitude (and quite low).

Table R1. Surface areas for LaPrO₃, LaPrO_{3+x}@3Li₂CO₃, LaPrO_{3+x}@5Li₂CO₃ and LaPrO_{3+x}@10Li₂CO₃

Sample	LaPrO ₃	LaPrO _{3+x} @3Li ₂ CO ₃	LaPrO _{3+x} @5Li ₂ CO ₃	LaPrO _{3+x} @10Li ₂ CO ₃
Surface area (m ² /g)	0.559	1.732	1.801	1.953

In order to confirm the surface carbonates under Raman, we have first conducted an *in-situ* Raman experiment for LaPrO_{3+x}@5Li₂CO₃ under 5% CO₂/Ar and ramped up the temperature from 120 to 700°C. As shown in Fig. R2(a), LaPrO_{3+x}@5Li₂CO₃ exhibited a clear surface carbonate peak between 1100-1300 cm⁻¹. This peak, however, tends to be broadened and smoothed out when the temperature gradually ramped up to 700°C. Note that bulk Li₂CO₃ has a melting point of 723°C, the broadening effect could be due to the melting of surface Li₂CO₃, which forms more amorphous phase and covers the LaPrO_{3+x} bulk in the form of thinner layers. We note that this broadening effect is less likely due to the thermal decomposition of Li₂CO₃. According to HSC Chemistry 9,

the decomposition of Li_2CO_3 via $\text{Li}_2\text{CO}_3 = \text{Li}_2\text{O} + \text{CO}_2(\text{g})$ has a ΔG of 75.62 kJ/mol and an equilibrium constant of $8.7\text{E-}5$. As such, this reaction is not thermodynamically favored in the presence of 5 Vol.% CO_2 . We have also compared *ex-situ* Raman under air in room temperature for $\text{LaPrO}_{3+x}@3\text{Li}_2\text{CO}_3$, $\text{LaPrO}_{3+x}@5\text{Li}_2\text{CO}_3$ and $\text{LaPrO}_{3+x}@10\text{Li}_2\text{CO}_3$. As can be seen Fig. R2(b), all these samples exhibited surface carbonate peaks of similar relative intensities. Thus, the absence of surface carbonate peaks for $\text{LaPrO}_{3+x}@5\text{Li}_2\text{CO}_3$ under *in-situ* Raman is more likely due to a temperature effect rather than Li_2CO_3 loading effect. We have added these discussions into the manuscript and added these figures below into the supplemental section.

Figure R2. (a) *In-situ* Raman for $\text{LaPrO}_{3+x}@5\text{Li}_2\text{CO}_3$ under 5% CO_2/Ar ; (b) *Ex-situ* Raman for $\text{LaPrO}_{3+x}@3\text{Li}_2\text{CO}_3$, $\text{LaPrO}_{3+x}@5\text{Li}_2\text{CO}_3$ and $\text{LaPrO}_{3+x}@10\text{Li}_2\text{CO}_3$ under air in room temperature.

3. Fig. 5c presents a comparison of $\text{LaPrO}_{3+x}@5\text{Li}_2\text{CO}_3$ with previously reported OCM catalysts. The authors claimed that $\text{LaPrO}_{3+x}@5\text{Li}_2\text{CO}_3$ exhibits the highest OCM yield reported to date and is the only catalyst surpassing the 30% single-pass C_{2+} yield at 100% methane partial pressure. Additionally, they highlighted that the optimal operating temperature for $\text{LaPrO}_{3+x}@5\text{Li}_2\text{CO}_3$ (700°C) is lower than that of classical OCM catalysts like $\text{Mn-Na}_2\text{WO}_4/\text{SiO}_2$ (~850°C). However, it may not be appropriate to directly compare the performance of $\text{LaPrO}_{3+x}@5\text{Li}_2\text{CO}_3$ in a CL-OCM experiment to that of a $\text{Mn-Na}_2\text{WO}_4/\text{SiO}_2$ catalyst in a steady-state experiment. The authors should test their material in steady state experiments to compare with the $\text{Mn-Na}_2\text{WO}_4/\text{SiO}_2$ catalyst or alternatively, test the $\text{Mn-Na}_2\text{WO}_4/\text{SiO}_2$ catalyst in a CL-OCM experiment.

Response: We thank the reviewer for the suggestion. We have synthesized and tested the $\text{Mn-Na}_2\text{WO}_4/\text{SiO}_2$ under redox OCM to compare with the reported $\text{LaPrO}_{3+x}@5\text{Li}_2\text{CO}_3$ in the manuscript. Three reaction temperatures were used, namely 700, 750 and 800°C. Methane partial pressure was set at 0.4 atm and GHSV was selected as 1050 h^{-1} , consistent with Point c in the figure below. At 700°C, $\text{Mn-Na}_2\text{WO}_4/\text{SiO}_2$ can only achieved C_{2+} yield of 12.38%. Increasing the temperature would increase the C_{2+} yield, and 18.57% yield was obtained at 800°C. However, this is still much lower than the reported $\text{LaPrO}_{3+x}@5\text{Li}_2\text{CO}_3$, which achieved 25.40% C_{2+} yield at the same gas flow conditions and 100°C lower (700°C). The redox OCM experiments were added into Fig. 5c in the manuscript, shown as Point e, f and g in the figure (shown as Fig. R3 below).

Figure R3. Revised Fig. 5c with added data points for Mn-Na₂WO₄/SiO₂ under redox OCM conditions

4. The authors found that the hydroxyl radical is highly active for methane activation. However, the authors didn't further discuss the impact of hydroxyl radical on CH₃ dimerization or decomposition on LaPrO_{3+x}@5Li₂CO₃, which could significantly influence the yield of C₂₊ hydrocarbons.

Response: We appreciate the very helpful comment. Meanwhile, we respectfully submit that the OH radicals are formed on the gas-molten salt interface, and they are likely to only concentrate near the gas-molten salt interface and get consumed quickly. Therefore, it is far more likely to contribute to surface CH₃ radical initiation than CH₃ dimerization since the latter would involve multiple radical species.

On the other hand, we do agree that if extensive OH radicals are present in the gas phase, it would affect a range of gas phase reactions in addition to methane activation. A well-known reaction mechanism, namely Gri Mech 3.0¹, includes a total of 352 radical reactions involving 53 species. Given that the rate determining step is the methane activation on the gas-molten salt interface, we respectfully submit that detailed consideration of the gas phase radical reactions is beyond the scope of the current work.

5. The authors performed AIMD calculations using the Vienna ab initio Simulation package (VASP). However, they did not include details regarding the convergence criteria used for electronic minimization.

Response: Thanks for the comment. In the AIMD calculation, the convergence criteria of force and energy were set to 0.01 eV/Å and 10⁻⁵ eV respectively. We have added this information into the manuscript.

6. Figure 4(b-f) is difficult to read due to color combinations and small font size. In addition, the authors should clarify what the colored balls in the molecular model represent.

Response: We thank the reviewer for the suggestion. We have increased the font size and increased the figure size to make it easier to read. We have also clarified the representations for the colored balls. The revised figure is shown in Fig. R4 below.

Figure R4. Revised figure for Fig. 4 in the manuscript. The yellow balls represent the highlighted oxygen atoms observed in AIMD calculations.

Reviewer 2:

In this draft, the authors prepared Li₂CO₃-coated mixed rare earth oxides for OCM reaction. It was found that on LaPrO_{3+x}@5Li₂CO₃, one of the three best catalysts, the one-way C₂₊ yield of 30.6% can be achieved at 700 degree C with a chemical loop reaction strategy. The single pass yield of 30% is usually regarded as the minimum requirement to industrialize this interesting and important reaction. Up to date, on all the tried catalysts with a constant CH₄+O₂ feed, it has not accomplished this goal yet. From this point of view, I believe the results reported in this draft is interesting for fundamental study, though the GHSV to obtain this value is very low. With the combination of experimental and DFT calculation, the authors have analyzed the structures of the catalysts in detail and tried to figure out the factors leading to the high yield. They found that the surface layer of Li₂CO₃ is important to maintain the reaction performance of the catalysts, which can stabilize surface peroxide anions and Pr⁴⁺ cations, both are critical to OCM reaction on the catalysts prepared in this study.

Response: We thank the reviewer for the positive comments. Our responses to the specific comments are listed below.

The draft can be accepted after addressing the following issues:

1. It is well-known that Mn-Na₂WO₄/SiO₂ is the best catalyst formulation up to date. Did the authors compare its reaction performance with that of LaPrO_{3+x}@5Li₂CO₃ (or one of the best three catalysts prepared by the authors) under the same condition? Either using the chemical loop strategy or traditional fixed-bed set-up with constant CH₄+O₂ flow in lean O₂. Although the authors do a comparison in Figure 5(c), I believe that most of the reaction data were not obtained with a chemical loop method if they were reported in former publications. Thus, I suggest the authors test the reaction performance of Mn-Na₂WO₄/SiO₂ under the same condition for more clear information.

Response: We thank the reviewer for the suggestion. We have synthesized and tested the Mn-Na₂WO₄/SiO₂ under redox OCM to compare with the reported LaPrO_{3+x}@5Li₂CO₃ in the manuscript. Three reaction temperatures were used, namely 700, 750 and 800°C. Methane partial pressure was set at 0.4 atm and GHSV was selected as 1050 h⁻¹, consistent with Point c in the figure below. At 700°C, Mn-Na₂WO₄/SiO₂ can only achieved C₂₊ yield of 12.38%. Increasing the temperature would increase the C₂₊ yield, and 18.57% yield was obtained at 800°C. However, this is still much lower than the reported LaPrO_{3+x}@5Li₂CO₃, which achieved 25.40% C₂₊ yield at the same gas flow conditions at a much lower temperature of 700°C. The redox OCM experiments were added into Fig. 5c in the manuscript, shown as Point e, f and g in the figure (shown as Fig. R5 below).

Figure R5. Revised Fig. 5c with added data points for Mn-Na₂WO₄/SiO₂ under redox OCM conditions

2. As to the title, it could be better to change into “Lithium carbonate-promoted....”. The current title will obviously lead people’s thought into the formation of active surface carbonate intermediate. But this is not main point. Moreover, 5% is not a low loading for Li₂CO₃, but the authors did not detect it with Raman. Since Raman is very sensitive to micro-crystallites, and the surface area of the sample is very low, this is unusual. In Figure S4, the authors in fact see carbonates with In-situ DRIFTS even in 10% O₂, can this carbonate peak be assigned to Li₂CO₃? If so, then how to explain the increase of carbonate peak after introducing CH₄. As I understand, part of Li₂O is still present in the shell layer.

Response: Thanks for the comment. We have changed the title per your suggestion.

In order to confirm the surface carbonates under Raman, we have first conducted an *in-situ* Raman experiment for $\text{LaPrO}_{3+x}@5\text{Li}_2\text{CO}_3$ under 5% CO_2/Ar and ramped up the temperature from 120 to 700°C. As shown in Fig. R2(a), $\text{LaPrO}_{3+x}@5\text{Li}_2\text{CO}_3$ exhibited a clear surface carbonate peak between 1100-1300 cm^{-1} . This peak, however, tends to be broadened and smoothed out when the temperature gradually ramps up to 700°C. Note that bulk Li_2CO_3 has a melting point of 723°C, the broadening effect could be due to the melting of surface Li_2CO_3 , which forms more amorphous phase and covers the LaPrO_{3+x} bulk in the form of thinner layers. We note that this broadening effect is less likely due to the thermal decomposition of Li_2CO_3 . According to HSC Chemistry 9, the decomposition of Li_2CO_3 via $\text{Li}_2\text{CO}_3 = \text{Li}_2\text{O} + \text{CO}_2(\text{g})$ has a ΔG of 75.62 kJ/mol and an equilibrium constant of $8.7\text{E-}5$. As such, this reaction is not thermodynamically favored in the presence of 5 Vol.% CO_2 . We have also compared *ex-situ* Raman under air in room temperature for $\text{LaPrO}_{3+x}@3\text{Li}_2\text{CO}_3$, $\text{LaPrO}_{3+x}@5\text{Li}_2\text{CO}_3$ and $\text{LaPrO}_{3+x}@10\text{Li}_2\text{CO}_3$. As can be seen Fig. R2(b), all these samples exhibited surface carbonate peaks of similar relative intensities. Thus, the absence of surface carbonate peaks for $\text{LaPrO}_{3+x}@5\text{Li}_2\text{CO}_3$ under *in-situ* Raman is more likely due to a temperature effect rather than Li_2CO_3 loading effect.

Figure R6. (a) *In-situ* Raman for $\text{LaPrO}_{3+x}@5\text{Li}_2\text{CO}_3$ under 5% CO_2/Ar ; (b) *Ex-situ* Raman for $\text{LaPrO}_{3+x}@3\text{Li}_2\text{CO}_3$, $\text{LaPrO}_{3+x}@5\text{Li}_2\text{CO}_3$ and $\text{LaPrO}_{3+x}@10\text{Li}_2\text{CO}_3$ under air in room temperature.

We agree with the reviewer that under reactions conditions at 700°C, a portion of the Li_2CO_3 could thermally decompose and form Li_2O . The increase of carbonate peak in DRIFTS-FTIR after introducing CH_4 may be assigned to the re-combination of the as-decomposed Li_2O and CO_2 , which is a by-product from OCM. We agree that part of Li_2O is still present in the shell layer. In fact, the presence of Li_2O in the shell layer should be beneficial as it can combine with electrophilic oxygen species on the LaPrO_{3+x} surface and form Li_2O_2 . We have added the above figures into the supplemental session and added these discussions in the manuscript.

3. During the chemical loop process, in the step of CH_4 injection, is there any carbon deposits formed. Though the authors have demonstrated that during 50 cycles, the catalyst is regenerable by O_2 and stable, some coking being detrimental to the catalyst structure could still occur after long-term running. It will be interesting to clarify this issue.

Response: We appreciate the comment. To confirm whether coking is present after long-term running, we have conducted another redox cycle on the sample after long-term run and uses a mass

spectrometer to track the product evolution. The results are shown in Fig. R7. In the figure below, Mass 15, Mass 26, Mass 30, Mass 32, and Mass 44 represents the characteristics peaks of CH₄, ethylene, ethane, O₂ and CO₂ respectively. We note that Mass 28 can both represent a characteristic peak of CO, or a fragment peak of ethylene or ethane. As can be seen, CO and CO₂ were not observed during the re-oxidation step, indicating that the coke formation was negligible. The abrupt signal change at the end of the OCM step is due to gas/valve switching.

Figure R7. Mass spectrometer spectrum for an OCM redox cycle on LaPrO₃@5Li₂CO₃ after long-term cycles

4. Some typos: line 187, “Fig. 4a” should be “Fig. 3a”, line 218, “Fig. 4c” should be “Fig. 3c”.

Response: We are sorry for the typos. These have been fixed in the revised manuscript.

Reviewer 3:

This an interesting and insightful paper, discussing the reaction mechanism of chemical looping experiments for oxidative coupling of methane, using new LaPrO₃-based catalysts. Overall, the results and the conclusions from the results are very exciting. My only comments are about details in the study and the role of Li₂CO₃ vs Li₂O₂. Please see below.

Response: We thank the reviewer for the positive comments. The responses to the specific comments are listed below.

1. Abstract

Can the authors give specific information about the “practical methane partial pressures”?

Response: We appreciate the comment. It was reported that most OCM studies were carried out under low methane partial pressures (i.e. well below 1 atm).² In practical industrial operations, methane partial pressure of at or above atmospheric pressure would be needed.³

2. Does a catalyst system mean a catalyst composite? Or a novel catalytic scheme for the process?

Response: We thank for the comment. We actually intended to mean a family of catalysts. As the reviewer has pointed out, the use of the word “system” is inappropriate. We have revised them in the manuscript.

3. How does a 30.6% yield compare to previous studies and industrially competitive processes?

Response: We appreciate the comment. Typical OCM catalysts include Li/MgO and Mn-Na₂WO₄/SiO₂. For Li/MgO, C₂₊ yield of ~20% can be obtained at temperatures around 700-750°C.⁴⁻⁷ Mn-Na₂WO₄/SiO₂ can obtain higher C₂₊ yields, albeit at higher temperatures (> 800°C). However, the C₂₊ yields have not exceeded 30%. We have also synthesized Mn-Na₂WO₄/SiO₂ in house and tested it using our redox conditions at temperatures between 700 and 800°C. At 700°C, Mn-Na₂WO₄/SiO₂ can only achieved C₂₊ yield of 12.38%. Increasing the temperature would increase the C₂₊ yield, and 18.57% yield was obtained at 800°C. However, this is still much lower than the reported LaPrO_{3+x}@5Li₂CO₃ in this manuscript. We have plotted these points into Fig. 5c in our manuscript to make a comparison on the C₂₊ yield and reaction temperature.

In terms of industrial competitiveness, we note that a previous process analysis study indicated that OCM can only be commercially viable with over 30-35% C₂₊ yield.⁸⁻¹⁰ Therefore, the current study is a step towards the right direction to make OCM competitive.

4. Is Pr essential in the catalytic composite – if yes, why not describe it as the best catalyst among those analyzed? Then the section about the Pr oxidation state will be better connected to the rest of the abstract.

Response: We thank for the comment. As discussed in the introduction, the use of Pr-containing mixed metal oxide is originated from the work by Gaffney et al., which reported C₂₊ yield of 16% at 775°C on a Na-impregnated Pr₆O₁₁.¹¹ We agree with the reviewer that it would be more appropriate to discuss the necessity of having Pr in the mixed metal oxide. However, we note that pure Pr₆O₁₁ promoted with carbonates do not perform well. We have also synthesized and tested LaCeO_{3+x}@5Li₂CO₃ and LaNdO_{3+x}@5Li₂CO₃, with Ce and Nd located very close to Pr in the periodic table in the lanthanide family. As shown in Fig. R8 for the redox OCM performance, both LaCeO_{3+x}@5Li₂CO₃ and LaNdO_{3+x}@5Li₂CO₃ exhibited very low C₂₊ yields and high selectivities towards CO₂. The necessity of Pr in the mixed metal oxide is probably due to the unique redox pair of Pr⁴⁺↔Pr³⁺, which leads to efficient generation of peroxide oxygen species in the Li₂CO₃ salt. We have added these discussions and added the figure below in the supplemental section.

Figure R8. Redox OCM performance comparison for $\text{LaPrO}_{3+x}@5\text{Li}_2\text{CO}_3$, $\text{LaCeO}_{3+x}@5\text{Li}_2\text{CO}_3$ and $\text{LaNdO}_{3+x}@5\text{Li}_2\text{CO}_3$. Temperature = 700°C, P_{CH_4} = 0.4 atm, GHSV = 1050 h⁻¹.

5. What is this optimization strategy (composition, structure, operability state)? Can you give a short description?

Response: We appreciate the reviewer for the insightful question. The optimization of an OCM catalyst has been a long-lasting and challenging task. Generally speaking, the endeavor has been pointed to two directions: (1) increasing the catalyst activity by enhancing methane activation and (2) decreasing CO_x selectivity by suppressing unselective oxidation of CH₃ radical or secondary oxidation of C₂₊.^{3,12} In this work, we established a $\text{LnPrO}_{3+x}@5\text{Li}_2\text{CO}_3$ redox catalyst family and the relationship between the oxidation state of Pr in the mixed oxide and the catalyst performance. LnPrO_{3+x} can provide Pr⁴⁺ and interact with the surface Li₂CO₃ salt. Peroxide oxygen species are then created and retained in the salt, leading to the formation of OH radical and more feasible methane activation pathways. Moreover, the Li₂CO₃ salt could cover the surface of LnPrO_{3+x} and block unselective sites, leading to decreased CO_x selectivity. This novel mechanism also leads to decreased operating temperatures around 700°C.

6. Introduction

Please remove and/or in the first paragraph. Again, I object to the use of the word “system” when describing multi-component catalysts. Please, reconsider for this manuscript.

Response: Thanks for pointing this out. Indeed, the use of the word “system” is inappropriate. We have revised them in the manuscript.

7. Previous work listed – please clarify whether the experiments were conducted continuously.

Response: We appreciate the comment. Typical OCM catalysts include Li/MgO and Mn-Na₂WO₄/SiO₂. These catalysts were generally conducted under a continuous, or a O₂-cofeed mode. We have added these into our revised introduction. Redox OCM have also been attempted. Typical catalysts include Na promoted Pr₆O₁₁ and Li/W co-doped Mg₆MnO₈. We have clarified this for each of these catalysts in the revised manuscript.

8. Green et al. and Labinger et al.’s modelling studies – please reveal why the C₂₊ yield cannot exceed 30-35%.

Response: Thanks for pointing this out. Labinger et al. solved a set of ordinary differential equations (ODE) derived via a pseudo-elementary reaction mechanism to ultimately chart methane conversion vs C₂ selectivity. In this model, the surface irreversibly reacts with CH₄, CH₃, C₂H₆, C₂H₄, and C₃₊ species. Initial rate parameter estimates for his mechanism are derived from experimental data for a typical mixed Mn–Mg oxide catalyst. By manipulating select rate constants to advantageous values consistent with other experimental catalytic data, Labinger found that an upper bound of ~30% yield at 1 atm methane partial pressure is present for redox OCM.³ Green et al. further considered the effects of transport and O₂ presence in an O₂-cofeed OCM. By maximizing all desired reaction rates and optimizing thermochemistry for all surface species on an ideal catalyst, Green solved that the C₂₊ yield is still limited to ~28% in the O₂-cofeed OCM.¹³ Thus, unless a radically new catalyst with fundamentally different reaction pathways is discovered, the C₂₊ yield cannot exceed 30-35% based on these modeling studies. We have added some of these discussions into the revised manuscript.

9. The authors say: “As such, a gap clearly exists between reported academic research results and industrial application” What are the requirements for industrial application?

Response: Thanks for the comment. It is commonly reported in the literatures that a C_{2+} yield of 30–35 % is needed to make OCM commercially viable.^{8–10} We have added this in the revised manuscript.

10. “spectroscopy (XPS) spectrometer” something is missing here, perhaps, “using a” spectrometer?

Response: We are sorry for the typos. We have made correction in the revised manuscript.

11. “is isolated, pseudotetrahedral Na-coordinated WO_4 surface sites” – consider correcting to “are isolated pseudotetrahedral Na-coordinated WO_4 surface sites.”

Response: We are sorry for the typos. We have made correction in the revised manuscript.

12. Results

“all the Li_2CO_3 promoted $LnPrO_{3+x}$ oxides ($Ln = La, Eu, Ho, Dy, Sm, Nd$) were active for OCM” – where can we see this?

Response: We are sorry for the confusion. These results were actually placed at the end of the result session to support our correlations between the Pr oxidation states of the support and C_{2+} yield. We think that this placement might benefit the flow of the manuscript. We have changed this sentence to “all the Li_2CO_3 promoted $LnPrO_{3+x}$ oxides ($Ln = La, Eu, Ho, Dy, Sm, Nd$) were active for OCM, as will be discussed in later sections” to avoid this confusion.

13. Fig. 1 b – is EELS used in anything for this result – you only show EDX mapping, or did I misunderstand? Please confirm whether the EDX (or EELS) image is a cross-section of a particle or an external surface of a particle.

Response: We are sorry for the confusion. Fig. 1b is the result of the EELS, which determines the Li element surface distribution. It is generally accepted that EELS can detect light elements such as Li, and EDX is more suitable for the detection of heavier elements. As such, we used TEM-EDX to determine the surface distribution of C element, and the results are shown in Fig. S1.

14. What benefits are of using in-situ TEM in Fig. 1e over ex-situ?

Response: We appreciate the comment. We think that the benefits of using *in-situ* TEM in Fig. 1e is that we can probe the material bulk and surface structures under high temperatures. The OCM uses reaction temperature of $700^\circ C$, close to the melting temperature of Li_2CO_3 ($723^\circ C$). We discovered under *in-situ* TEM that the surface Li_2CO_3 stays as an amorphous phase and uniformly covers the $LaPrO_{3+x}$ bulk at $700^\circ C$.

15. Working with low loading of carbonates is usually associated with problems confirming the presence of the carbonate. The XPS results confirm the presence of Li_2CO_3 , but can the authors comment on the phase diagram of this component – is it expected to react or decompose under CH_4 or O_2 ? Where does the peroxide species come from if it is associated with the carbonate? What happens to carbon during the creation of Li_2O_2 ? Is the carbonate even needed in cycling, or rather the goal is to have Li_2O_2 (or Li_2O) all along? Have you tried depositing Li_2O and oxidizing it to obtain Li_2O_2 directly?

Response: We appreciate the comment. Indeed, it is a tricky answer whether Li_2CO_3 is maintained on the surface or not. Based on the *in-situ* XPS (Fig. 1c) and *in-situ* TEM (Fig. 1e) results in the manuscript, we think that Li_2CO_3 is still present on the surface. However, this indeed does not exclude that Li_2CO_3 may partly decompose into Li_2O . Based on HSC Chemistry 9, the decomposition of Li_2CO_3 via $\text{Li}_2\text{CO}_3 = \text{Li}_2\text{O} + \text{CO}_2(\text{g})$ has a ΔG of 75.62 kJ/mol and an equilibrium constant of 8.729E-005. Although the ΔG is not high, this reaction could happen in atmospheres without CO_2 . Thus, in O_2 atmosphere, partial decomposition of Li_2CO_3 can happen and this is consistent with our *in-situ* DRIFTS-FTIR experiments (Fig. S5) where carbonate peak decreases during the O_2 re-oxidation step. During the OCM step, we note that CO_2 is actually a key product with a selectivity of ~20%. Thus, there is appreciable amount of CO_2 partial pressure during the OCM step, and this could lead to the recombination of Li_2O and CO_2 into Li_2CO_3 . This is also consistent with our *in-situ* DRIFTS-FTIR experiments in Fig. S5, where the carbonate peak becomes more pronounced during the OCM step.

Despite the analysis above, we agree with the reviewer that during any stage of the reaction under high temperatures, the surface should be in a mixed state of both Li_2O and Li_2CO_3 . We actually think that the presence of Li_2O inside the Li_2CO_3 could be beneficial, as Li_2O can easily accept the active oxygen species on the LaPrO_{3+x} surface and form Li_2O_2 . The as-formed Li_2O_2 can then act as a “peroxide carrier” and transport the active oxygen species to the outer layer to react with methane. The reviewer also raised a very good point of directly depositing Li_2O onto the surface. We simulated that by using LiNO_3 rather than Li_2CO_3 for wet impregnation onto LaPrO_{3+x} with the same Li loading. After sintering, the surface would be more dominant with Li_2O rather than Li_2CO_3 . After several redox cycles, we discovered that $\text{LaPrO}_{3+x}@\text{LiNO}_3$ also exhibited activity for OCM, although the C_{2+} yield is lower than that of $\text{LaPrO}_{3+x}@\text{Li}_2\text{CO}_3$ (Fig. R9). Again, we note that the Li_2O on the LaPrO_{3+x} could partly transform into Li_2CO_3 due to recombination of Li_2O and by-product CO_2 . The lower activity and especially high CO_2 selectivity of $\text{LaPrO}_{3+x}@\text{LiNO}_3$ could be due to the smaller amount of Li_2CO_3 on the surface, which does not effectively retain peroxide inside the salt and does not entirely cover the unselective sites on LaPrO_{3+x} surface.

Figure R9. Redox OCM performance comparison of sintered $\text{LaPrO}_{3+x}@\text{Li}_2\text{CO}_3$ and $\text{LaPrO}_{3+x}@\text{LiNO}_3$ using the same Li loading. Temperature = 700°C, $P_{\text{CH}_4} = 0.4$ atm, GHSV = 1050 h^{-1} .

16. Fig. 3 – can you identify the oxygen in the carbonate species in O1s for the in-situ XPS (I see it in the SI for ex-situ experiments)?

Response: We thank the reviewer for the catch. We have deconvoluted the O 1s peaks. From Fig. R10(a) shown below, the carbonate O peak is still present after the OCM step. And the peak located between 531 to 532 eV during the re-oxidation step should be assigned to a combination of both carbonate O and peroxide O. We have revised this in the manuscript.

Indeed, we note that the carbonate species peak in the *ex-situ* XPS is much higher than *in-situ* XPS. We think that this could be due to two reasons: (1) there are more CO₂ absorbed onto the sample surface at *ex-situ* conditions due largely to the low operating pressure of *in-situ* XPS, and (2) there are more bulk Li₂CO₃ at room temperature while more surface and amorphous Li₂CO₃ are present at high temperatures. Both could lead to a higher carbonate O peak in the *ex-situ* XPS. For the second point, we have conducted an *in-situ* Raman experiment for LaPrO_{3+x}@5Li₂CO₃ under 5%CO₂/Ar and ramped up the temperature from 120 to 700°C for further confirmation. Detailed response can be referred to our answer to Reviewer 1's Question 2.

Figure R10. Deconvolution of O 1s peaks for the *in-situ* XPS results

17. Is the effect of cooperation between Li₂CO₃ and LaPrO_{3+x} dependent on the extent of the interface or on the molar or mass ratios? Can the authors comment based on their extensive experimental results?

Response: We thank for the reviewer for the comment. Please refer to our answer to Reviewer 1's Question 1. Briefly speaking, the molar or mass ratio between the carbonate and LaPrO_{3+x} does have an impact on the catalyst performance. We found an optimal formulation being LaPrO_{3+x}@5Li₂CO₃ although both lower and higher loading of Li₂CO₃ would be beneficial for OCM when compared to unpromoted LaPrO_{3+x}.

18. I fail to see any meaningful result in Fig. 4 d – the caption says the figure shows the relative intensity for OH radical – in what sense relative? What is on the x, y axes?

Response: We are sorry for the confusion and the miss. In the LIF experiment, we used a ceramic boat to contain our samples and applied an OH-sensitive camera at one side to capture the induced OH radical signal by the laser. We have added the detailed experimental information into the supplemental section. The x, y axes in our Fig. 4d is the dimension of our capture image in the units of cm. We have also modified the color bar and color scale. In the revised figure, we set the highest OH radical signal using the most intense color in the color bar, and normalized the relative signal

intensity from 0 to 1 in the color bar. The revised figure is shown in Figure R4 in the response to Question 6 in Reviewer 1.

19. Catalyst reactivity – please comment on the longevity of catalysts other than LaPrO₃@5Li₂CO₃.

Response: We appreciate the comment. Typical OCM catalysts include Li/MgO and Mn-Na₂WO₄/SiO₂. For Li/MgO, it was reported that the deactivation is mainly due to a loss of Li as LiOH.¹⁴ For example, Korf et al. reported that the C₂ yield dropped from 14% to 5% within 20 hrs at 800°C with Li/MgO (ca. 3.1 wt.% Li).¹⁵ It was also reported that the deactivation could be partly suppressed by transforming Li₂O into the more stable Li₂CO₃.¹⁴ As compared, Mn-Na₂WO₄/SiO₂ is generally considered as a stable catalyst. For example, Li et al. tested Mn-Na₂WO₄/SiO₂ in a fluidized bed for 450 h time on stream at a temperature of 800–875°C and observed relatively constant performance. They also reported fixed-bed tests with 500 and 1000h on stream and reported stable performance.¹⁶ We have added these discussions into the revised manuscript.

20. How do the results (mainly product distribution) depend on the time of reduction and oxidation? I do not consider this essential for the paper, as the paper focuses on comparing different materials, but I am curious whether this can be an important factor – so I hope the authors can comment on their decision for the time steps in cycling.

Response: We thank for the comment. Indeed, the reduction time could play a role in the redox OCM, as basically similar in any other related chemical looping approaches that utilizes lattice oxygen. We have tested redox OCM on LaPrO_{3+x}@5Li₂CO₃ using reduction time of 30, 60 and 90 s, while holding the oxidation time of 180 s. As can be seen in Fig. R11, smaller reduction time would lead to increased methane conversion but decreased C₂₊ selectivity, probably due to the more facile release of more unselective oxygen at the beginning. Larger reduction time led to decreased methane conversion but increased C₂₊ selectivity, probably due to the consumption of lattice oxygen with extended reduction time. Overall, the highest C₂₊ yield was obtained at reduction time of 60 s. We also found that the oxidation time is not as important if the time period is long enough to completely regenerate the lattice oxygen. A longer oxidation time would not hurt the redox catalyst in general. We note that this conversion and selectivity trend is also consistent with other chemical looping and redox OCM investigations.^{17,18}

Figure R11. Redox OCM performance on $\text{LaPrO}_{3+x}@5\text{Li}_2\text{CO}_3$ using reduction time of 30, 60 and 90 s. Temperature = 700°C , $P_{\text{CH}_4} = 0.4 \text{ atm}$, $\text{GHSV} = 1050 \text{ h}^{-1}$.

21. The authors wrote: “In addition to Li_2CO_3 , other alkali metal carbonates such as Na_2CO_3 and K_2CO_3 were also investigated using LaPrO_{3+x} as the redox-active component”. Can you add information on the yields of C_2^+ achieved, or even add that information to Fig. 6?

Response: We appreciate the comment. We have added the redox OCM performance results of $\text{LaPrO}_{3+x}@5\text{Na}_2\text{CO}_3$ and $\text{LaPrO}_{3+x}@5\text{K}_2\text{CO}_3$, and added it alongside Fig. 6 as Fig. 6(b). As can be seen in Fig. R12, switching from Li_2CO_3 to Na_2CO_3 and K_2CO_3 leads to decreased catalyst activities and decreased C_2^+ yields. This might be due to the lower activities of Na_2O_2 and K_2O_2 than Li_2O_2 , where Na_2O_2 and K_2O_2 are more stable than Li_2O_2 according to HSC Chemistry 9.0 as seen in Table R2.

Figure R12. Redox OCM performance comparison for LaPrO_{3+x} , $\text{LaPrO}_{3+x}@5\text{Li}_2\text{CO}_3$, $\text{LaPrO}_{3+x}@5\text{Na}_2\text{CO}_3$ and $\text{LaPrO}_{3+x}@5\text{K}_2\text{CO}_3$. Temperature = 700°C , $P_{\text{CH}_4} = 0.4 \text{ atm}$, $\text{GHSV} = 1050 \text{ h}^{-1}$.

Table R2. Standard Gibbs free energy change for decomposition of Li_2O_2 , Na_2O_2 and K_2O_2 at 700°C

Reactions	Standard ΔG at 700°C
$2\text{Li}_2\text{O}_2 = 2\text{Li}_2\text{O} + \text{O}_2(\text{g})$	-90.206 kJ/mol
$2\text{Na}_2\text{O}_2 = 2\text{Na}_2\text{O} + \text{O}_2(\text{g})$	45.239 kJ/mol
$2\text{K}_2\text{O}_2 = 2\text{K}_2\text{O} + \text{O}_2(\text{g})$	105.604 kJ/mol

22. Methods

Are the results of the catalyst OCM activity from the start of the cycle, the end, or the average across the cycle?

Response: Thanks for the comment. The results of the catalyst OCM activity are obtained based on the average values across the cycle. We have added this description in Methods: Reactivity test session.

23. How were selectivity, conversion and yield calculated?

Response: The gaseous products were collected in a gas bag and injected into a gas chromatograph (GC) for analysis. The GC was an Agilent 7890 Fast RGA with a flame ionization detector (FID) for hydrocarbon analysis and two thermal conductivity detectors for CO, CO₂, and H₂ analysis. A refinery gas standard (Agilent Part # 5190-0519) was used to calibrate the GC. Thus, the selectivity, conversion and yield were calculated based on the average gas product in the gas bag, which contained all gas outlets from the 60 s reduction step. The equations used for calculating are listed below. We have added these in the method session.

$$\text{Methane Conversion} = \frac{\text{Methane Input} - \text{Methane Output}}{\text{Methane Input}}$$

$$\text{C}_{2+} \text{ Selectivity} = \frac{\text{moles of C in C}_{2+} \text{ products}}{\text{moles of C in converted methane}}$$

$$\text{C}_{2+} \text{ Yield} = \text{Methane Conversion} * \text{Selectivity of C}_{2+}$$

24. Supplementary

Fig. S6 and Fig. S9 – please add information about the experimental temperature. Can you comment on whether the oxidation step brings the bare LaPrO_{3+x} mass to the original value?

Response: We are sorry for the oversight. We have added the experimental temperature both in the paragraph and on the figure. We have also normalized the absolute weight change to the change of weight percentage (100%) to make it clearer. The weight percentage change of LaPrO_{3+x}@5Li₂CO₃ is shown in Fig. R13(a). As can be seen, the weight percentage change during redox cycles is 0.8 wt.% and oxidation step brings the mass of LaPrO_{3+x}@5Li₂CO₃ to its original value. We have also conducted 20 redox cycles on bare LaPrO_{3+x}, as shown in Fig. R13(b). As can be seen, the weight percentage change during redox cycles is ~1.6 wt.% and oxidation step also brings the mass of LaPrO_{3+x} to its original value. This is consistent with O₂-TPD that Li₂CO₃ has decreased the reducibility of LaPrO_{3+x}.

Figure R13. Catalyst weight changes during redox OCM cycles for $\text{LaPrO}_{3+x}@5\text{Li}_2\text{CO}_3$ and LaPrO_3 at 700°C

References

- (1) Ianni, J. C. - A Comparison of the Bader-Deuflhard and the Cash-Karp Runge-Kutta Integrators for the GRI-MECH 3.0 Model Based on the Chemical Kinetics Code Kintecus. In *Computational Fluid and Solid Mechanics 2003*; Bathe, K. J., Ed.; Elsevier Science Ltd: Oxford, 2003; pp 1368–1372. <https://doi.org/10.1016/B978-008044046-0.50335-3>.
- (2) Gao, Y.; Neal, L.; Ding, D.; Wu, W.; Baroi, C.; Gaffney, A. M.; Li, F. Recent Advances in Intensified Ethylene Production—A Review. *ACS Catal.* **2019**, *9* (9), 8592–8621. <https://doi.org/10.1021/acscatal.9b02922>.
- (3) Labinger, J. A. Oxidative Coupling of Methane: An Inherent Limit to Selectivity? *Catal. Lett.* **1988**, *1* (11), 371–375. <https://doi.org/10.1007/BF00766166>.
- (4) Ito, T.; Wang, J.; Lin, C. H.; Lunsford, J. H. Oxidative Dimerization of Methane over a Lithium-Promoted Magnesium Oxide Catalyst. *J. Am. Chem. Soc.* **1985**, *107* (18), 5062–5068.
- (5) Ito, T.; Lunsford, J. H. Synthesis of Ethylene and Ethane by Partial Oxidation of Methane over Lithium-Doped Magnesium Oxide. *Nature* **1985**, *314* (6013), 721–722. <https://doi.org/10.1038/314721b0>.
- (6) Luo, L.; You, R.; Liu, Y.; Yang, J.; Zhu, Y.; Wen, W.; Pan, Y.; Qi, F.; Huang, W. Gas-Phase Reaction Network of Li/MgO-Catalyzed Oxidative Coupling of Methane and Oxidative Dehydrogenation of Ethane. *ACS Catal.* **2019**, *9* (3), 2514–2520.
- (7) Arndt, S.; Simon, U.; Heitz, S.; Berthold, A.; Beck, B.; Görke, O.; Epping, J.-D.; Otremba, T.; Aksu, Y.; Irran, E.; Laugel, G.; Driess, M.; Schubert, H.; Schomäcker, R. Li-Doped MgO From Different Preparative Routes for the Oxidative Coupling of Methane. *Top. Catal.* **2011**, *54* (16), 1266. <https://doi.org/10.1007/s11244-011-9749-z>.
- (8) Kondratenko, E. V.; Peppel, T.; Seeburg, D.; Kondratenko, V. A.; Kalevaru, N.; Martin, A.; Wohlrab, S. Methane Conversion into Different Hydrocarbons or Oxygenates: Current Status and Future Perspectives in Catalyst Development and Reactor Operation. *Catal. Sci. Technol.* **2017**, *7* (2), 366–381. <https://doi.org/10.1039/C6CY01879C>.
- (9) Vandewalle, L. A.; Van de Vijver, R.; Van Geem, K. M.; Marin, G. B. The Role of Mass and Heat Transfer in the Design of Novel Reactors for Oxidative Coupling of Methane. *Chem. Eng. Sci.* **2019**, *198*, 268–289. <https://doi.org/10.1016/j.ces.2018.09.022>.
- (10) Jašo, S.; Godini, H. R.; Arellano-Garcia, H.; Omidkhah, M.; Wozny, G. Analysis of Attainable Reactor Performance for the Oxidative Methane Coupling Process. *Chem. Eng. Sci.* **2010**, *65* (24), 6341–6352.

- (11) Gaffney, A. M.; Jones, C. A.; Leonard, J. J.; Sofranko, J. A. Oxidative Coupling of Methane over Sodium Promoted Praseodymium Oxide. *J. Catal.* **1988**, *114* (2), 422–432.
- (12) Gambo, Y.; Jalil, A. A.; Triwahyono, S.; Abdulrasheed, A. A. Recent Advances and Future Prospect in Catalysts for Oxidative Coupling of Methane to Ethylene: A Review. *J. Ind. Eng. Chem.* **2018**, *59*, 218–229. <https://doi.org/10.1016/j.jiec.2017.10.027>.
- (13) San Su, Y.; Ying, J. Y.; Green Jr, W. H. Upper Bound on the Yield for Oxidative Coupling of Methane. *J. Catal.* **2003**, *218* (2), 321–333.
- (14) Arndt, S.; Laugel, G.; Levchenko, S.; Horn, R.; Baerns, M.; Scheffler, M.; Schlögl, R.; Schomäcker, R. A Critical Assessment of Li/MgO-Based Catalysts for the Oxidative Coupling of Methane. *Catal. Rev.* **2011**, *53* (4), 424–514. <https://doi.org/10.1080/01614940.2011.613330>.
- (15) Korf, S. J.; Roos, J. A.; de Bruijn, N. A.; van Ommen, J. G.; Ross, J. R. H. Oxidative Coupling of Methane over Lithium Doped Magnesium Oxide Catalysts. *Catal. Today* **1988**, *2* (5), 535–545. [https://doi.org/10.1016/0920-5861\(88\)85017-X](https://doi.org/10.1016/0920-5861(88)85017-X).
- (16) Arndt, S.; Otremba, T.; Simon, U.; Yildiz, M.; Schubert, H.; Schomäcker, R. Mn–Na₂WO₄/SiO₂ as Catalyst for the Oxidative Coupling of Methane. What Is Really Known? *Appl. Catal. Gen.* **2012**, *425–426*, 53–61. <https://doi.org/10.1016/j.apcata.2012.02.046>.
- (17) Gao, Y.; Neal, L. M.; Li, F. Li-Promoted LaSr₂-XFeO_{4-δ} Core–Shell Redox Catalysts for Oxidative Dehydrogenation of Ethane under a Cyclic Redox Scheme. *ACS Catal.* **2016**, *6* (11), 7293–7302. <https://doi.org/10.1021/acscatal.6b01399>.
- (18) Baser, D. S.; Cheng, Z.; Fan, J. A.; Fan, L.-S. Codoping Mg-Mn Based Oxygen Carrier with Lithium and Tungsten for Enhanced C₂ Yield in a Chemical Looping Oxidative Coupling of Methane System. *ACS Sustain. Chem. Eng.* **2021**, *9* (7), 2651–2660.

REVIEWER COMMENTS

Reviewer #1 (Remarks to the Author):

The authors have significantly improved the manuscript. I just have one remaining suggestion for revision. After the authors have made changes in response to this minor comment, the manuscript can be accepted.

The authors have updated Figure 4 and provided clarification for the atom representations. However, in the revised manuscript, the authors have not specified the meaning of the yellow-highlighted atoms. Do they represent the oxygen atoms observed in AIMD calculations? This clarification should be included in the revised manuscript.

Reviewer #2 (Remarks to the Author):

The revised draft has been carefully read by this reviewer. All the questions proposed by this reviewer have been addressed and clarified by the authors, even with some additional experiments. The quality of the draft has been improved. I thus recommend accept it for publication.

Reviewer #3 (Remarks to the Author):

The authors replied to my questions with detail and attention - much appreciated. May reiterate one of the missed questions, please - can you explain whether the sample analysed with EELS was a cross-section or an external surface of a particle? How was the sample prepared for EELS?

Response to comments from reviewers:

Reviewer 1:

The authors have significantly improved the manuscript. I just have one remaining suggestion for revision. After the authors have made changes in response to this minor comment, the manuscript can be accepted.

The authors have updated Figure 4 and provided clarification for the atom representations. However, in the revised manuscript, the authors have not specified the meaning of the yellow-highlighted atoms. Do they represent the oxygen atoms observed in AIMD calculations? This clarification should be included in the revised manuscript.

Response: We thank the reviewer for the positive comments.

We apologize for the lack of clarity and have updated the diagram. Specifically, the electrophilic oxygens that are directly involved in the reactions are highlighted in Figure 4b and 4c. The highlighted oxygen atoms are intended to guide the readers to better visualize the configurational changes. For example, with the highlighted oxygen atoms in Figure 4b, one can see that an H_2O_2 is decomposed into OOH^- after the AIMD simulation. We have also changed the viewing angle for improved visualization. Additional descriptions are added in the figure captions for clarifications.

Figure 4. (a) Summary of the possible reaction product of $\text{H}_2\text{O} + \text{O}_2^{2-}$; (b) and (c): Mean energies as a function of elapsed time ($t-t_0$) for evolution of $\text{H}_2\text{O}_2 + \text{O}^{2-}$ and $\text{OH}^- + \text{CO}_4^{2-}$ in molten Li_2CO_3 , respectively. The electrophilic oxygen atoms that are involved in the reactions are highlighted in yellow to provide better visualization; (d) LIF experiments on $\text{SiO}_2@5\text{Li}_2\text{CO}_3$, scale bar shows the relative intensity for OH radical; (e) and (f): Mean reaction energies as a function of elapsed time ($t-t_0$) for evolution of $\text{CH}_4 + \text{O}_2^{2-}$ and $\text{CH}_4 + \text{OH}^*$ over molten Li_2CO_3 surface, respectively.

Reviewer 2:

The revised draft has been carefully read by this reviewer. All the questions proposed by this reviewer have been addressed and clarified by the authors, even with some additional experiments. The quality of the draft has been improved. I thus recommend accept it for publication.

Response: We thank the reviewer for the positive comments.

Reviewer 3:

The authors replied to my questions with detail and attention - much appreciated. May reiterate one of the missed questions, please - can you explain whether the sample analyzed with EELS was a cross-section or an external surface of a particle? How was the sample prepared for EELS?

Response: We thank the reviewer for the positive comments. In our EELS, we observed the external surface of the particle, not the cross-section. The sample preparation procedure for EELS is provided as following:

“The TEM sample was prepared by dry casting the oxide particles on a carbon film supported by copper mesh grid. The sample mesh grid was then inserted into and TEM sample chamber and was vacuumed. EELS mapping was done on over the entire particle. On the edge of the particle, EELS signal mostly comes from the external surface.”

We have clarified in the revised manuscript and added these additional details in the experimental session of the supplemental file.

REVIEWERS' COMMENTS

Reviewer #1 (Remarks to the Author):

The revised version is now acceptable for publication.

Reviewer #3 (Remarks to the Author):

Thank you for the revision. The paper looks great.

Response to comments from reviewers:

Reviewer 1:

The revised version is now acceptable for publication.

Response: We thank the reviewer for the positive comments.

Reviewer 3:

Thank you for the revision. The paper looks great.

Response: We thank the reviewer for the positive comments.